# Efficient Online Reinforcement Learning Fine-Tuning Need Not Retain Offline Data

**Zhiyuan Zhou**[*1]**, Andy Peng**[*1]**, Qiyang Li**[1]**, Sergey Levine**[1]**, Aviral Kumar**[2]
[1]UC Berkeley, [2]Carnegie Mellon University     ([*]Equal Contribution)

## Abstract

The modern paradigm in machine learning involves pre-training on diverse data, followed by task-specific fine-tuning. In reinforcement learning (RL), this translates to learning via offline RL on a diverse historical dataset, followed by rapid online RL fine-tuning using interaction data. Most RL fine-tuning methods require continued training on offline data for stability and performance. However, this is undesirable because training on diverse offline data is slow and expensive for large datasets, and should, in principle, also limit the performance improvement possible because of constraints or pessimism on offline data. In this paper, we show that retaining offline data is unnecessary as long as we use a properly-designed online RL approach for fine-tuning offline RL initializations. To build this approach, we start by analyzing the role of retaining offline data in online fine-tuning. We find that continued training on offline data is mostly useful for preventing a sudden divergence in the value function at the onset of fine-tuning, caused by a distribution mismatch between the offline data and online rollouts. This divergence typically results in unlearning and forgetting the benefits of offline pre-training. Our approach, Warm-start RL (WSRL), mitigates the catastrophic forgetting of pre-trained initializations using a very simple idea. WSRL employs a warmup phase that seeds the online RL run with a very small number of rollouts from the pre-trained policy to do fast online RL. The data collected during warmup bridges the distribution mismatch, and helps "recalibrate" the offline Q-function to the online distribution, allowing us to completely discard offline data without destabilizing the online RL fine-tuning. We show that WSRL is able to fine-tune without retaining any offline data, and is able to learn faster and attains higher performance than existing algorithms irrespective of whether they do or do not retain offline data.

## 1 Introduction

The predominant paradigm for learning at scale today involves pre-training models on diverse prior data, and then fine-tuning them on narrower domain-specific data to specialize them to particular downstream tasks (Devlin et al., 2018; Brown et al., 2020; Driess et al., 2023; Radford et al., 2021; Zhai et al., 2023; Touvron et al., 2023; Zhou et al., 2024). In the context of learning decision-making policies, this paradigm translates to pre-training on a large amount of previously collected static experience via offline reinforcement learning (RL), followed by fine-tuning these initializations via online RL efficiently. Generally, this fine-tuning is done by continuing training with the very same offline RL algorithm, e.g., pessimistic (Kumar et al., 2020; Cheng et al., 2022) algorithms or algorithms that apply behavioral constraints (Fujimoto and Gu, 2021; Kostrikov et al., 2021), on a mixture of offline data and autonomous online data, with minor modifications to the offline RL algorithm itself (Nakamoto et al., 2024).

While this paradigm has led to promising results (Kostrikov et al., 2021; Nakamoto et al., 2024), RL fine-tuning requires continued training on offline data for stability and performance ((Zhang et al., 2023; 2024); Section 3), as opposed to the standard practice in machine learning. Retaining offline data is problematic for several reasons. First, as offline datasets grow in size and diversity, continued online training on offline data becomes inefficient and expensive, and such computation requirements may even deter practitioners from using online RL for fine-tuning. Second, the need for retaining offline data perhaps defeats the point of offline RL pre-training altogether: recent results (Song et al., 2023), corroborated by our experiments in Section 3, indicate that current fine-

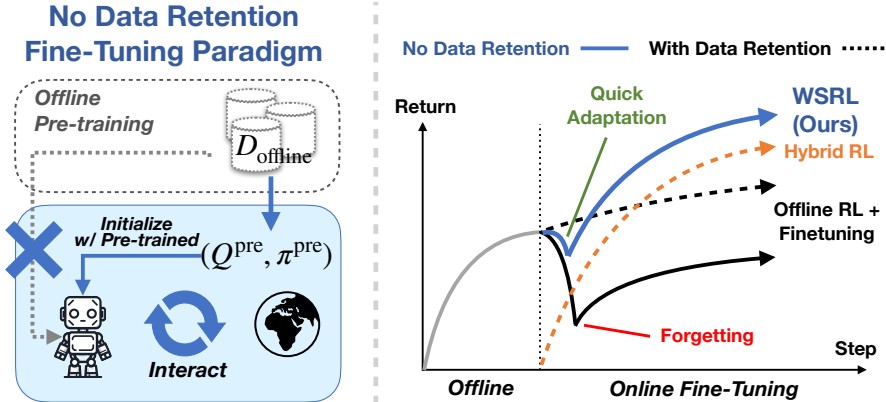

Figure 1: **No data retention fine-tuning** focuses on RL fine-tuning without using the offline dataset during online updates, mirroring the common paradigm in machine learning at scale today. The offline dataset is only used to pre-train a policy and Q-function via offline RL to initialize fine-tuning, after which the dataset is discarded and the agent only fine-tunes with online experience. Current methods struggle in this "no-retention" setting and **forget** knowledge learned from pre-training. Our goal is to develop a fine-tuning method that **quickly adapts** online even if we **do not retain offline data**.

tuning approaches are not able to make good use of several strong offline RL value and/or policy initializations, as shown by the superior performance of running online RL from scratch with offline data put in the replay buffer (Ball et al., 2023). These problems put the efficacy of current RL fine-tuning approaches into question.

In this paper, we aim to understand and address the aforementioned shortcomings of current online fine-tuning methods and build an online RL approach that does *not* retain offline data. To develop our approach, we first empirically analyze the importance of retaining offline data in current online RL fine-tuning algorithms. We find that for both pessimistic (e.g., CQL (Kumar et al., 2020)) and behavioral constraint (e.g., IQL (Kostrikov et al., 2021)) algorithms, the offline Q-function undergoes a "recalibration" phase at the onset of online fine-tuning where its values change substantially. This recalibration phase can lead to unlearning and even complete forgetting of the offline initialization. This manifests in the form of divergent value functions when no offline data is present for training. Even methods specifically designed for fine-tuning (e.g. CalQL (Nakamoto et al., 2024)) still suffer from this problem with limited or no offline data. This extends the observation of Nakamoto et al. (2024) about unlearning in the standard offline-to-online fine-tuning setting in that it demonstrates that forgetting and unlearning pose a more severe challenge in no data retention fine-tuning.

*Is it possible to fine-tune from offline RL value and policy initializations, but without retaining offline data and not forget the pre-training?* Our key insight is that seeding online fine-tuning with even a small amount of appropriately collected data that "simulates" offline data retention, but *more in distribution* to the online fine-tuning task, can greatly facilitate recalibration, mitigating the distribution mismatch between pre-training and fine-tuning and preventing forgetting. Once this recalibration is over, we can run the most effective online RL approach (without pessimism or behavioral constraints) for most sample-efficient online learning. Our approach, WSRL (**W**arm **S**tart **R**einforcement **L**earning), instantiates this idea by incorporating a warmup phase to initialize the online replay buffer with a small number of online rollouts from the pre-trained policy, and then running the best online RL method with various offline RL initializations to fine-tune. WSRL is able to learn faster and attains higher asymptotic performance than existing algorithms irrespective of whether they retain offline data or not. This approach is not a particularly novel or clever algorithm, but it perhaps is one of the more natural approaches to enable effective fine-tuning of offline initializations.

Our main contribution in this paper is the study of RL online fine-tuning with no offline data retention, a paradigm we call *no-retention fine-tuning*. We provide a detailed analysis of existing offline-to-online RL methods and find that offline data is often needed during fine-tuning to mitigate the Q-value divergence and the resulting forgetting due to distribution shift, but also slows down fine-tuning asymptotically. We demonstrate that a simple method of incorporating a warmup phase to initialize the replay buffer with a small number of transitions from the pre-trained offline RL pol-

icy followed by running a simple online RL algorithm is effective at sample-efficient fine-tuning, without forgetting the pre-trained initialization.

## 2 PROBLEM FORMULATION: FINE-TUNING WITHOUT OFFLINE DATA

We operate in an infinite-horizon Markov Decision Process (MDP), $\mathcal{M} = \{\mathcal{S}, \mathcal{A}, \mathbf{P}, r, \gamma, \rho\}$, consisting of a state space $\mathcal{S}$, an action space $\mathcal{A}$, a transition dynamics function $\mathbf{P}(s'|s, a) : \mathcal{S} \times \mathcal{A} \mapsto \mathcal{P}(\mathcal{A})$, a reward function $r : \mathcal{S} \times \mathcal{A} \mapsto \mathbb{R}$, a discount factor $\gamma \in [0, 1)$, and an initial state distribution $\rho : \mathcal{P}(\mathcal{S})$. We have access to an offline RL pre-trained policy $\pi_\psi^{\text{pre}}(a|s) : \mathcal{S} \mapsto \mathcal{P}(\mathcal{A})$ and pre-trained Q-function $Q_\theta^{\text{pre}}(s, a) : \mathcal{S} \times \mathcal{A} \mapsto \mathbb{R}$. Our goal is to build an online fine-tuning algorithm that only uses $\pi_\psi^{\text{pre}}(a|s)$ and $Q_\theta^{\text{pre}}(s, a)$ and not $\mathcal{D}_{\text{off}}$ to maximize the discounted return: $\eta(\pi) = \mathbb{E}_{s_{t+1} \sim \mathbf{P}(\cdot|s_t, a_t), a_t \sim \pi(\cdot|s_t), s_0 \sim \rho} \sum_{t=0}^{\infty} [\gamma^t r(s_t, a_t)]$.

**Problem setup.** Note that the RL fine-tuning problem we study in this paper *does not* allow retaining $\mathcal{D}_{\text{off}}$. We will refer to this problem setting as ***no retention online fine-tuning***. Conceptually, our problem setting is close to the offline-to-online fine-tuning problem (Nair et al., 2020; Kostrikov et al., 2021; Nakamoto et al., 2024), but we do not allow data retention.

## 3 UNDERSTANDING THE ROLE OF OFFLINE DATA IN ONLINE FINE-TUNING

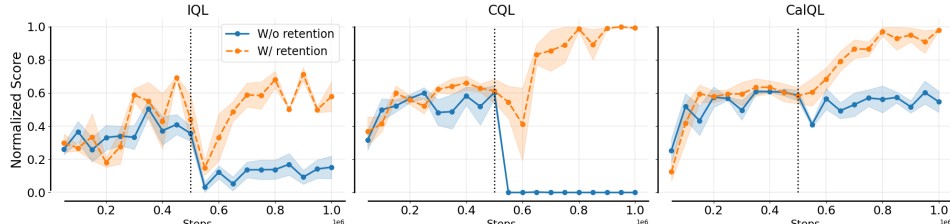

Figure 2: In no-retention fine-tuning, IQL, CQL, and CalQL all fail to fine-tune on `kitchen-partial`. In contrast, when continually training on offline data during fine-tuning, these algorithms work as intended. Vertical dotted line indicates the separation between pre-training and fine-tuning.

We first attempt to understand why current offline-to-online RL methods require retaining the offline dataset. In particular, we hope to understand the pros and cons of retaining offline data and gain insights into developing new methods that serve a similar role, but do not require retaining offline data. We center our study along two axes: **(1)** we analyze the role of retaining offline data at the beginning of fine-tuning, and **(2)** we analyze the effect of retaining offline data on asymptotic fine-tuning performance.

### 3.1 THE ROLE OF OFFLINE DATA AT THE BEGINNING OF FINE-TUNING

Extending observations from prior work (Nakamoto et al., 2024), we find that fine-tuning offline RL initializations fails severely if no offline data is retained. Specifically, observe in Figure 2 that offline RL algorithms IQL (Kostrikov et al., 2021) and CQL (Kumar et al., 2020) unlearn right at the beginning of fine-tuning, with performance dropping down to nearly a 0% success rate on the `kitchen-partial` task from D4RL (Fu et al., 2020b). More importantly, they are not able to recover over the course of fine-tuning. CalQL (Nakamoto et al., 2024), an offline RL approach specifically designed for subsequent fine-tuning by learning calibrated Q-functions, initially drops in performance, but improves with further online training. That said, it still struggles to improve beyond its pre-trained performance.

To better understand the above results, we introduce some terminology. We define **"unlearning"** as the performance drop at the start of fine-tuning, with possible recovery later, and **"forgetting"** as the destruction of pre-trained initialization at the beginning of fine-tuning such that recovery becomes nearly impossible with online RL training. In general, unlearning may be unavoidable due to the distribution shift over state-action pairs between offline and online (e.g. consider a sparse reward problem where minor change in the policy action results in huge change in return) (Xie et al., 2021). On the other hand, forgetting the pre-trained initialization altogether is problematic since it defeats the benefits of offline RL pre-training. Our goals is for the agent to quicky recover after unlearning, which relies on the effective use of pre-trained knowledge. Observe in Figure 2 that while all algorithms unlearn, CQL and IQL forget. While CalQL does not forget, it does not

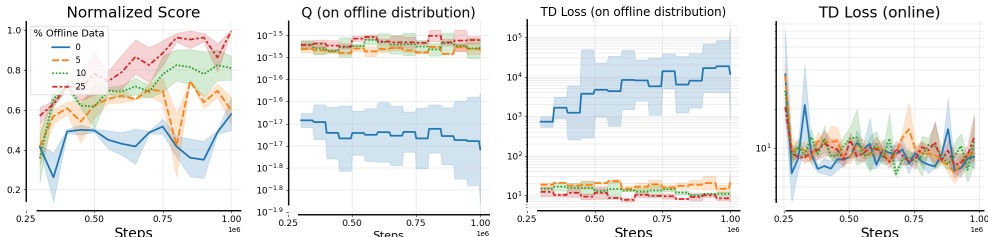

Figure 3: **When offline data is removed (to different extents) during fine-tuning**, performance drops (subfigure a) because Q-function fit on offline dataset distribution diverges (subfigure b, c), even though the Q-function can fit the online distribution (subfigure d). This plot shows fine-tuning CalQL on `kitchen-partial` with 0/5/10/25% offline data in each update batch. We have similar findings with IQL and CQL.

improve further. This indicates a bottleneck in fine-tuning with online RL without offline data, and different offline RL initializations suffer from this challenge to different extents.

***Why does not retaining offline data hurt?*** To build intuition of what can go wrong, we observe in Figure 3 what happens when retaining different amounts of offline data. Figure 3(b) shows that the average Q-value under the offline distribution begins to diverge as the amount of retained offline data decreases. This Q-value divergence, in turn, corresponds to a divergence in the TD-error (Figure 3(c)), perhaps highlighting forgetting. We will show in Section 5 how such divergence can be prevented by our method.

Diving deeper, we find that this divergence only appears under the distribution of the offline data (on which we evaluate metrics but do not train): TD-error on online distribution remain small regardless of the amount of offline data retained (Figure 3(d)); on the other hand, the TD error under the offline data distribution grows substantially with a decrease in offline data (Figure 3(c)). This trend is consistent across CQL, IQL, and CalQL, though the divergence for CalQL is the least severe perhaps thanks to its calibrated Q-function, which correlates with the stability and best performance of CalQL in this problem setting in Figure 2. This suggests that the problem with no data retention in current offline-to-online fine-tuning algorithms likely stems from a form of ***distribution shift*** between the online and offline data distribution. Fine-tuning on more on-policy data destroys how well we fit offline data. As we see in Figure 2, this can lead to forgetting of the pre-trained initialization.

> **Takeaway 1: Distribution shift between offline and online data destroys Q-function fit**
>
> Training only on online experience without offline data retention can destroy how well the model fits offline data: despite attaining comparable TD-errors on the online data to the setting when offline data is retained, TD-errors under the offline distribution grow larger.

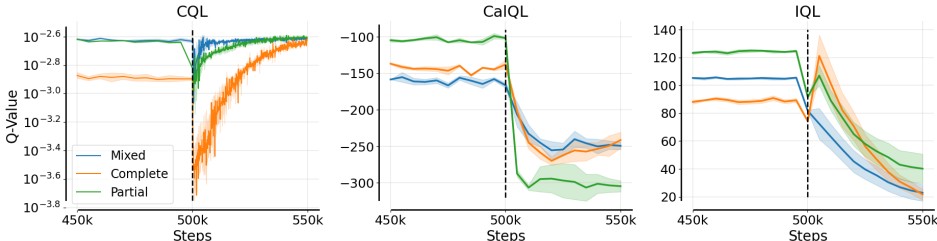

Figure 4: **A downward spiral effect** in CQL (left), CalQL (middle), and IQL (right) Q-functions in no-retention fine-tuning on `kitchen-mixed`, `kitchen-complete`, and `kitchen-partial`: When fine-tuning starts at 500k steps, Q function goes on a downward spiral. When it eventually recovers, the policy has already unlearned (Figure 2).

***Why are Q-values underestimated?*** Not only do the Q-values under the offline data distribution diverge, Q-values on the online distribution also go through underestimation at the onset of fine-tuning (see Figure 4). This Q-value divergence is a manifestation of the "***recalibration***" process at the boundary between offline RL and online fine-tuning, previously identified in Figure 3 of Nakamoto et al. (2024). However, unlike Nakamoto et al. (2024), the recalibration process in no retention fine-tuning must operate entirely on limited on-policy data. Thus, we see that despite

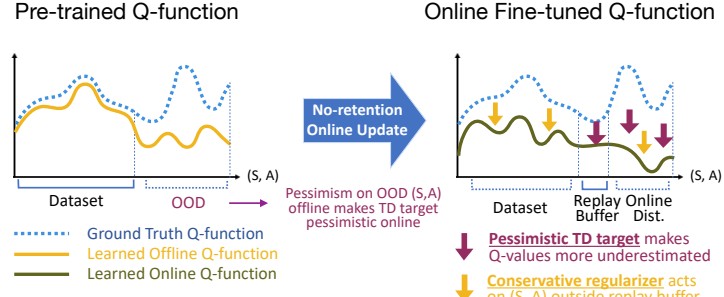

Figure 6: Retaining offline data is not efficient, and is outperformed by online RL methods like RLPD on three different environments. RLPD starts from scratch, and CalQL starts from pre-trained initialization at step 0.

CalQL explicitly modifies the scale of the offline Q-function initialization, it still cannot prevent forgetting when offline data is not retained.

Next, we wish to understand why recalibration leads to divergent Q-values. Figure 5 provides an intuition for such Q-value underestimation. Consider the very first batch of online rollouts collected from the environment. The target value computation for estimating TD-error on online state-action pairs will query the offline Q-function, $Q_\theta^{\text{pre}}$, on out-of-distribution state-action pairs. Due to conservative offline RL pre-training (e.g. CQL or CalQL), learned Q-values at these out-of-distribution state-action pairs are expected to be small. Using such pessimistic values for TD targets in the Bellman backup will, in turn, propagate these *underestimation* errors onto the new online state-action Figure 3(d) corroborates effective propagation of these TD targets. In addition, if one continues to run pessimistic RL algorithms during fine-tuning (e.g., CQL or CalQL), the conservative regularizer continues to minimize out-of-distribution Q-values: with few on-policy rollouts, Q-values for unseen actions keep getting smaller.

Figure 5: Illustration to demonstrate why Q-values are under-estimated in no-retention fine-tuning and may lead to a **"downward spiral"**.

This mechanistic understanding hints at a ***"downward spiral"*** in the learned Q-function at the beginning of fine-tuning. By this point the Q-function recovers, the policy has forgotten its pre-training and is no longer able to recover to its offline performance (Figure 2). We find that this phenomenon becomes more detrimental as the amount of offline data in fine-tuning is reduced, as shown in Figure 3. The most adverse effects arise when no offline data can be retained. Even calibrated algorithms, such as CalQL, though more robust, still suffer from this issue (Figure 4 middle).

> **Takeaway 2: Re-calibration of Q-values leads to excessive underestimation**
>
> We find that Q-value recalibration at the beginning of fine-tuning leads to excessive underestimation due to backups with over-pessimistic TD-targets. We call this the "downward spiral".

## 3.2 THE ADVERSE IMPACT OF OFFLINE DATA ON ASYMPTOTIC PERFORMANCE

As shown above, offline data plays an important role in current offline-to-online algorithms by helping with recalibration and preventing Q-value divergence. But how does it affect performance in the long term, once recalibration is over? We find that continued training on offline data *hurts* final performance and efficiency. Specifically, we find in Figure 6 that offline RL fine-tuning tends to be substantially slower than online RL algorithms from scratch that initialize the replay buffer with offline data (Ball et al., 2023; Song et al., 2023). This is quite concerning because it indicates that either offline RL pre-training provides no benefits for fine-tuning (unlike other fields of machine learning where pre-training helps substantially) or that existing RL fine-tuning approaches from various offline RL initializations are not effective enough to make use of offline initializations. We will show that a simple modification to online RL methods in the high updates-to-data (UTD) regime is able to make good use of initializations from several offline RL methods, without offline data.

> **Takeaway 3: Retaining offline data hurts asymptotic performance**
>
> While retaining offline data appears to be crucial for preventing forgetting at the beginning of fine-tuning for current fine-tuning methods, continuing to make updates on this offline data with an (pessimistic) offline RL algorithm negatively impacts asymptotic performance and efficiency.

## 4 WSRL: FAST FINE-TUNING WITHOUT OFFLINE DATA RETENTION

So far we saw that retaining offline data in offline RL algorithms can slow down online fine-tuning but we also cannot remove offline data due to forgetting. How can we tackle *both* the forgetting of offline initialization and attain asymptotic sample efficiency online?

**Key idea.** Perhaps one straightforward approach to address asymptotic efficiency issues is to utilize a standard online RL approach, with no pessimism or constraints for fine-tuning, unlike current offline-to-online fine-tuning approaches that still retain offline RL specific techniques in fine-tuning. We can further accelerate online learning by operating in the high updates-to-data (UTD) regime (Ball et al., 2023; Chen et al., 2021). The remaining question is: how do we tackle catastrophic forgetting at the onset of fine-tuning that prevents further improvements online, without offline data? Our insight is that we can "simulate" continued training on offline data by collecting a small number of *warmup* transitions with a *frozen* offline RL policy at the onset of online fine-tuning. Training on these transitions via an aggressive, high updates-to-data (UTD) online RL approach, without retaining offline data can mitigate the challenges of catastrophic forgetting. Our approach, WSRL (**W**arm **S**tart **R**einforcement **L**earning) instantiates these insights into an extremely simple and practical method that enables us to obtain strong fine-tuning results without offline data.

**WSRL algorithm.** WSRL is an off-policy actor-critic algorithm (Algorithm 1). It initializes the value function and policy with the pre-trained Q-function $Q_\theta^{\text{pre}}$ and policy $\pi_\psi^{\text{pre}}$ could come from any offline RL algorithm. Then, WSRL uses the first $K$ online steps to collect a few rollouts using the frozen offline RL policy to simulate the retention of offline data. We refer to this phase as the **"warmup" phase**. After warmup data collection, WSRL uses standard temporal-difference (TD) updates and policy gradient. For fine-tuning, we fine-tune with a high updates-to-data (UTD) ratio (Fu et al., 2019; Chen et al., 2021) and follow other best practices. To combat issues such as overestimation (Hasselt, 2010)in the high UTD regime, we use an ensemble of Q functions (Chen et al., 2021) and layer normalization (Hiraoka et al., 2022) in both the actor and the critic.

**Implementation details.** Most of the results in this paper use CalQL to initialize WSRL, even though in principle other initializations could be used. See Appendix H for running WSRL with different initializations. We choose Soft Actor-Critic (Haarnoja et al., 2018a), with an ensemble of 10 Q-networks and layer normalization, as our online fine-tuning algorithm. We use a UTD of $4$. This design is inspired by the work of Ball et al. (2023). We use $K = 5000$ warmup steps at the start of fine-tuning. Further implementation details are in Appendix I.

## 5 EXPERIMENTAL EVALUATION

The goal of our experiments is to study how well WSRL is able to fine-tune online without offline data retention. We also ablate the design decisions in WSRL to understand the efficacy of WSRL. Concretely, we study the following research questions: **(1)** Can WSRL enable efficient fine-tuning in the no-retention setting?; **(2)** How does WSRL compare with methods that do retain offline data?; **(3)** How critical is the warmup phase in WSRL?; **(4)** How important is it to use online RL algorithm for online fine-tuning?, and **(5)** How important is it to pre-train the policy, value function, or both?

### 5.1 BASELINES AND EXPERIMENTAL SETUP

While most prior RL fine-tuning methods are not designed explicitly for the no-retention fine-tuning setting, they can definitely be applied or repurposed to our setting. **JSRL** (Uchendu et al., 2023) uses a pre-trained policy to roll in for some number of steps during each episode and then rolls out with the current policy. The online policy is trained from scratch with both the roll-in and rollout experience. To improve JSRL's competitiveness, we also initialize the online policy with the pre-trained policy and run it with high UTD. Offline RL methods have also been shown to fine-tune online. We consider three offline methods, **CQL** (Kumar et al., 2020), **IQL** (Kostrikov et al., 2021), and **CalQL** (Nakamoto et al., 2024), an extension to CQL that calibrates the Q-values for efficient fine-tuning. Another method, **SO2** (Zhang et al., 2024) attempts to balance reward maximization

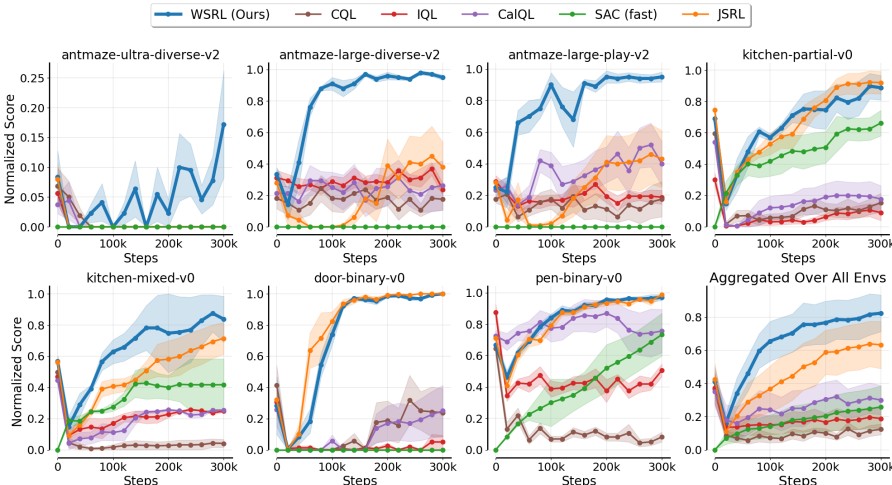

Figure 7: In **no-retention fine-tuning**, WSRL fine-tunes efficiently and greatly outperforms all previous algorithms, which often fail to recover from an initial dip in performance. JSRL, the closest baseline, uses a data-collection technique similar to warmup.

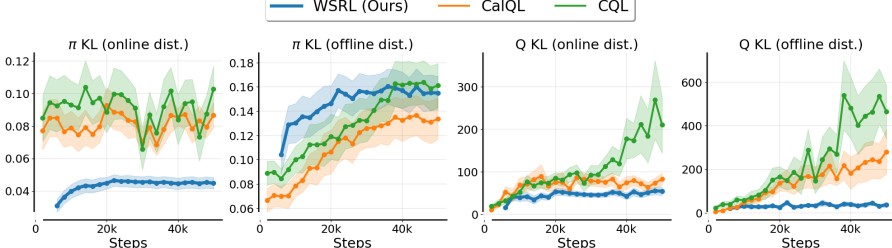

Figure 8: **KL divergence of the pre-trained policy and Q-function with fine-tuned ones** show that WSRL did not forget its pre-training. The four plots show $D_{\mathrm{KL}}(\pi_\psi^{\mathrm{pre}}||\pi_\psi)$ and $D_{\mathrm{KL}}(\mathrm{softmax}(Q_\theta^{\mathrm{pre}})||\mathrm{softmax}(\pi_\psi))$ on online and offline state distributions. This plot aggregates the KL divergence of six different environments shown separately in Figure 19 and 20, along with a more detailed discussion in Appendix E.

and pessimistic pre-training via high UTD and perturbed value updates during fine-tuning. Finally, **RLPD** (Ball et al., 2023) is an efficient online RL algorithm that learns from scratch by performing $50/50$ sampling of the offline dataset and online buffer for each update batch. While the typical fine-tuning recipe involves sampling each update batch from both the offline dataset and online replay buffer (CQL, CalQL, RLPD), or initializing the replay buffer with the offline dataset (IQL, SO2), we evaluate them in the no-retention setting by only sampling from the online buffer. Since RLPD *without* $50/50$ sampling is equivalent to a rapidly-updating Soft Actor Critic (Haarnoja et al., 2018a) agent, we refer to it as **SAC (fast)**.

**Experimental setup.** We study a number of challenging benchmarks tasks and pre-training dataset compositions following protocol used by prior works (Nakamoto et al., 2024; Kostrikov et al., 2021; Kumar et al., 2020). We experiment on Antmaze, Kitchen, and Adroit tasks from D4RL (Fu et al., 2020a) and the Gym MuJoCo locomotion tasks[1]. More discussion is in Appendix C.

### 5.2 CAN WSRL ENABLE EFFICIENT FINE-TUNING IN NO-RETENTION FINE-TUNING?

Figure 7 compares WSRL with the aforementioned methods applied to no-retention fine-tuning. In seven different tasks, WSRL significantly outperforms baselines, fine-tuning faster to a higher asymptotic performance. CQL, IQL, and CalQL completely fail in this setting, as noted before in Section 3. SAC (fast) completely fails in exploration-heavy environments, but can improve slowly in some environments. The most competitive baseline is JSRL. While JSRL achieves the same performance on `Adroit` as WSRL, it is significantly worse on `Antmaze` and `Kitchen`. We hypothesize that this performance gap stems from the ability of WSRL to benefit from the pre-trained value initialization, especially on datasets where the pre-trained Q-function is good (e.g. `Antmaze`). Note that while WSRL does suffer from an initial unlearning in performance (see Figure 18) it recovers quickly and outperforms other methods which often do not recover at all. As we noted

---

[1]Results in Appendix B.

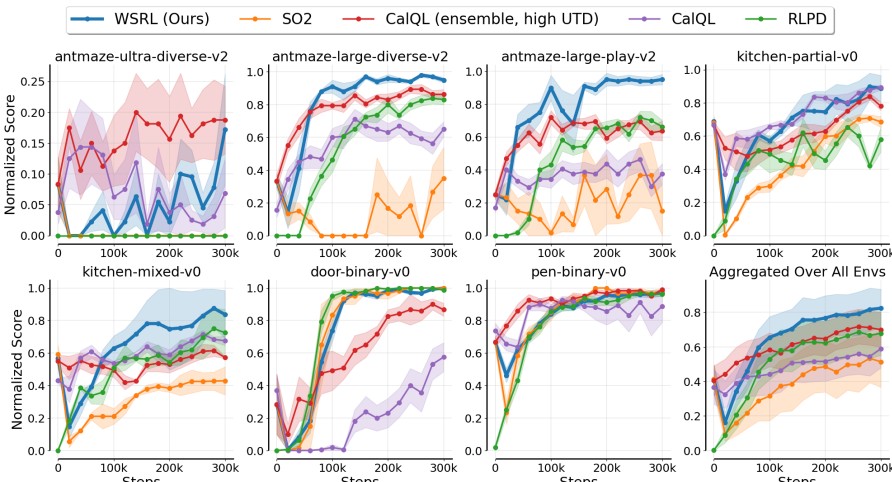

Figure 9: Compared to methods that **do retain offline data** online, WSRL, perhaps surprisingly, is still able to fine-tune faster or competitively despite not retaining any offline data. This highlights benefits of warmup.

before, some unlearning might be unavoidable in some environments, but it is important to prevent forgetting of the pre-trained initialization. To evaluate how much the policy and Q-function forget, we measure in Figure 8 the KL divergence between the pre-trained and fine-tuned Q-functions and separately between pre-trained and fine-tuned policies during the unlearning period. While CQL and CalQL suffer from divergence in the Q-function, WSRL's Q-function remains stable and relatively close to its pre-trained initialization. The KL divergence of the policy shows that $\pi_\psi$ does not forget its pre-training on states relevant in fine-tuning, while unlearning potentially irrelevant information. This shows that while WSRL unlearns initially, it does **not** forget. See Appendix E for details.

## 5.3   How does WSRL compare to methods that retain offline data?

In Figure 9, we compare WSRL to prior methods that still retain and utilize offline data during online fine-tuning. For example, the method labeled as CalQL in this figure would sample transitions from both the offline data and online data to construct an update batch. To make comparisons fair, we also compare to a version of all methods that trains with high UTD of 4 and an ensemble of 10 Q-functions. Observe that WSRL also outperforms these methods despite the fact that these prior methods retain the entire offline dataset during fine-tuning. Specifically, WSRL usually achieves higher asymptotic performance than CalQL and fine-tunes faster, indicating retaining offline data may not be the best compromise for asymptotic performance, as we have also shown in Section 3. WSRL also outperforms RLPD, indicating that WSRL can effectively utilize the pre-trained value function and policy initializations for rapid online learning.

## 5.4   How critical is the warmup phase?

We find that the warmup phase is crucial for fine-tuning with online RL. As shown in Figure 10, WSRL without warmup performs significantly worse in three different environments. Moreover, we find that using such simple warmup scheme is better than WSRL initializing with the same number of transitions from the offline dataset or retaining the offline dataset all together (See Appendix J and K). We hypothesize that warmup helps because it helps mitigate the distribution shift and prevents divergence and the "downward spiral" at the beginning of fine-tuning. As shown in Figure 11 shows, learned Q-values do not diverge to overly pessimistic values when warmup is employed and the TD losses remain small on the offline data as well, avoiding the "downward spiral" in Section 3. See more detailed discussion in Appendix G.

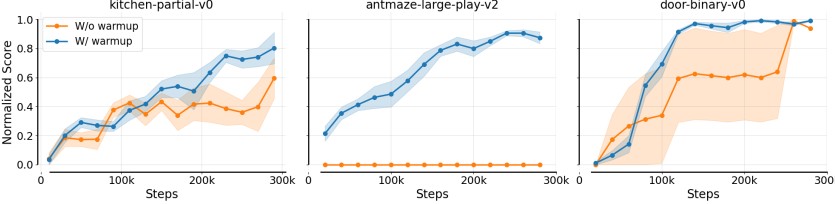

Figure 10: **Warmup is critical to fast fine-tuning:** When WSRL does not use the initial 5000 steps of warmup, it performs worse or has much higher variance.

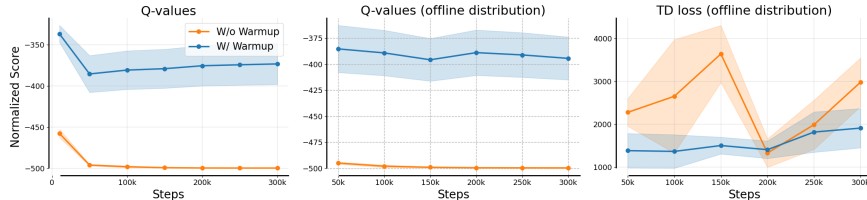

Figure 11: **Warmup phase helps prevent downward spiral**: (left) Q-values during fine-tuning, and warmup mitigates over-pessimistic values; (middle, right) Q-value and TD error evaluated on the offline distribution, where warmup prevents divergence. Data from WSRL on `Antmaze-large-play`.

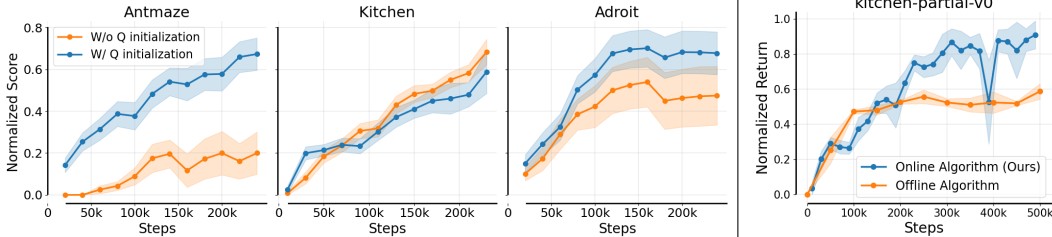

Figure 12: **Ablation studies:** the importance of using Q-function initialization (left) and a non pessimistic online RL algorithm (right). **Left: Q-function initialization** is especially helpful when the pre-training dataset has high coverage (e.g. `Antmazes`). Each plot averages across different dataset types on a domain; **Right:** Importance of fine-tuning with a standard **online RL algorithm**: SAC learns faster than CalQL.

## 5.5 How important is using a standard (non-pessimistic) online RL algorithm for fine-tuning?

In addition to a warmup phase, using a standard online RL algorithm for fine-tuning is also a critical design choice in WSRL. We ablate this decision by attempting to use an offline RL algorithm during fine-tuning. Here we choose to use CalQL, because it is less likely to experience Q-divergence (Section 3) by design, compared to CQL and IQL, but is still a pessimistic algorithm. We also initialize CalQL with the pre-trained policy and Q-function, and use an identical number of warmup steps online. As shown in Figure 12 (right), using an offline algorithm is significantly worse than using a standard online RL algorithm, SAC.

## 5.6 How important is it to initialize the policy, value function, and both?

**Importance of policy initialization.** At the start of fine-tuning, WSRL initializes the policy to the pre-trained policy. Since the pre-trained policy is already capable of meaningful actions, it speeds up online learning. In Figure 13, we compare WSRL's performance with and without "policy initialization", and find that initializing with the pre-trained policy is crucial for fast fine-tuning.

**Benefits of Q-value initialization.** In Figure 12 (left), we observe that while initializing the value function did not bring additional benefits in some domains, it made fine-tuning faster in others. For e.g., initializing with the Q-function is especially helpful in the `Antmaze` domains. We hypothesize that this is because the pre-training datasets in Antmazes

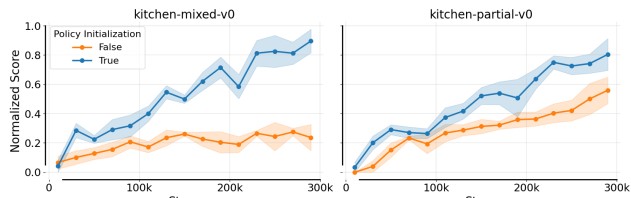

Figure 13: Importance of **policy initialization** in WSRL: with policy initialization, WSRL performs much better in `Kitchen`.

exhibit much broader coverage compared to those in `Adroit` and `Kitchen`, resulting in a better offline Q-function. Consequently, initializing with a more informative Q-function in Antmazes accelerates online fine-tuning.

## 6 Related Work

**Offline-to-online RL.** Offline-to-online RL focuses on leveraging an offline dataset to run online RL as sample-efficient as possible (Lee et al., 2022; Nair et al., 2020). Many methods developed for this setting utilize offline pre-training followed by a dedicated fine-tuning phase (Nair et al., 2020; Kostrikov et al., 2021; Agarwal et al., 2022; Hu et al., 2023; Rafailov et al., 2023; Nakamoto et al., 2024) *on a mix of offline and online data*. Offline RL methods can also be directly used for fine-tuning by continuing training when adding new online data to the offline data buffer (Kumar

et al., 2020; Kostrikov et al., 2021; Tarasov et al., 2024). Most similar to the goal in our paper is Agarwal et al. (2022), which attempts to use previous RL computations as a better initialization for downstream tasks. However, this work, along with all the methods above, still require retaining all of the pre-training data in the data buffer. As we also show, these methods completely fail without the offline data in the buffer. Our work does not retain offline data. Uchendu et al. (2023) utilizes a pre-trained policy to guide online fine-tuning without the need of offline data retention, but discard the value function, which typically drives learning in most actor-critic RL algorithms. Ji et al. (2023) and Luo et al. (2024) run offline RL and online RL concurrently on a shared replay buffer, following the idea of tandem learning (Ostrovski et al., 2021). Although the high-level motivating principle behind this line of work is also to use offline RL to boost online RL efficiency, there's no pre-training.

**Bottlenecks in online RL fine-tuning of offline RL policies.** In this work, we show that offline data retention greatly stabilizes the recalibration of the Q-function at the onset of fine-tuning, which otherwise can lead to catastrophic forgetting due to state-action distribution shift. Luo et al. (2023) observe that putting offline data into the offline RL replay buffer stabilizes fine-tuning, but also slows down learning. However, it still remains unclear as to why offline data hurts fine-tuning, which our analysis aims to answer. Lee et al. (2022) identify the existence of state-action distribution shift between offline data and online rollout data, but do not explicitly analyze the negative effects of this shift in online fine-tuning. Nakamoto et al. (2024) show the poor scale calibration of offline pre-trained Q-function to be a key cause for instability of pessimistic algorithms during online fine-tuning with offline data retention. Our analysis extends this analysis to the setting which does not retain offline data and uncovers a distinct reason (state-action distribution shift) that also plagues the method of Nakamoto et al. (2024) in this regime.

**Online RL with prior data but no pre-training.** Another line of work bypasses offline RL pre-training altogether, directly using a purely online RL agent to learn on data samples from both offline data and online interaction data from scratch (Song et al., 2022; Zhou et al., 2023; Ball et al., 2023). Despite not employing pre-training, evidence shows that this recipe can work pretty well, often outperforming online RL fine-tuning methods that utilize a separate offline pre-training phase. If the most effective way to utilize prior data is to include it in the replay buffer without any pre-training at all–no matter which pre-training algorithm is used–then it perhaps indicates that we are missing some important ingredients for a truly scalable RL formula for pre-training and fine-tuning. In this paper, we show that at least a big part of the problem lies in online fine-tuning of offline RL initializations, and build an extremely simple approach to fix the problem.

**Fine-tuning RL policies with no data retention.** Finally, many continual and lifelong RL methods also fine-tune policies without retaining prior experiences due to the non-stationarity assumption in the environment dynamics and task specification (Ring, 1994; Kirkpatrick et al., 2017; Huang et al., 2021; Wołczyk et al., 2021; Powers et al., 2022). Meta-RL methods (Duan et al., 2016; Rothfuss et al., 2018; Stadie et al., 2018; Rakelly et al., 2019; Arndt et al., 2020; Dorfman et al., 2021; Grigsby et al., 2023) assume access to a task/environment distribution to optimize for fast fine-tuning online. In contrast, we only consider the single-environment, single-task setting where the pretraining and fine-tuning are in the same environment for the same task. In the same single-environment, single task setting, many prior works study on-policy RL methods (e.g., PPO (Schulman et al., 2017)) to fine-tune pre-trained policies (Schaal, 1996; Kober and Peters, 2008; Rajeswaran et al., 2017; Gupta et al., 2019; Wołczyk et al., 2024; Ren et al., 2024). Among these, Wołczyk et al. (2024) also observe unlearning at the beginning of fine-tuning and find that explicitly mitigating unlearning with techniques from continual learning improves the efficiency of fine-tuning. In contrast to these works, we focus on off-policy actor-critic RL methods, that provide an elevated sample efficiency, and require different solution strategies to address this unlearning problem.

## 7 CONCLUSION

In this paper, we explore the possibility of fine-tuning RL agents online without retaining and co-training on any offline datasets. Such setting is important for truly scalable RL, where offline RL is used to pre-train on a diverse dataset, followed by online RL fine-tuning where keeping the offline data is expensive or impossible. We find that previous offline-to-online RL algorithms fail completely in this setting because of Q-value divergence due to distribution shift. However, if we simply use online RL algorithm for fine-tuning and allow the Q-values to stabilize through a warmup phase, we can prevent the Q-divergence. We hope that WSRL sheds light on the challenges in no-retention fine-tuning, and inspire future research on the important paradigm of no-retention RL fine-tuning.

## ACKNOWLEDGMENTS

We thank Seohong Park, Mitsuhiko Nakamoto, Kyle Stachowicz, and anonymous reviewers for informative discussions and feedback on an earlier version of this work. This research was supported by the AI Institute and Office of Naval Research under grants N00014-24-1-2206 and N00014-20-1-2383. We thank TPU Research Cloud (TRC) and Google Cloud for generous compute donations that made this work possible.

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

# Appendices

## A   ADDITIONAL RESULTS ON ANTMAZE ENVIRONMENTS

In the main paper, we presented results on three of the most challenging antmaze environments. Here, in addition to the set of three antmaze environments shown in Figures 7 and 9, we provide the results of WSRL on all eight D4RL antmaze environments, together with strong baseline methods. The results show that WSRL is significantly better than baselines.

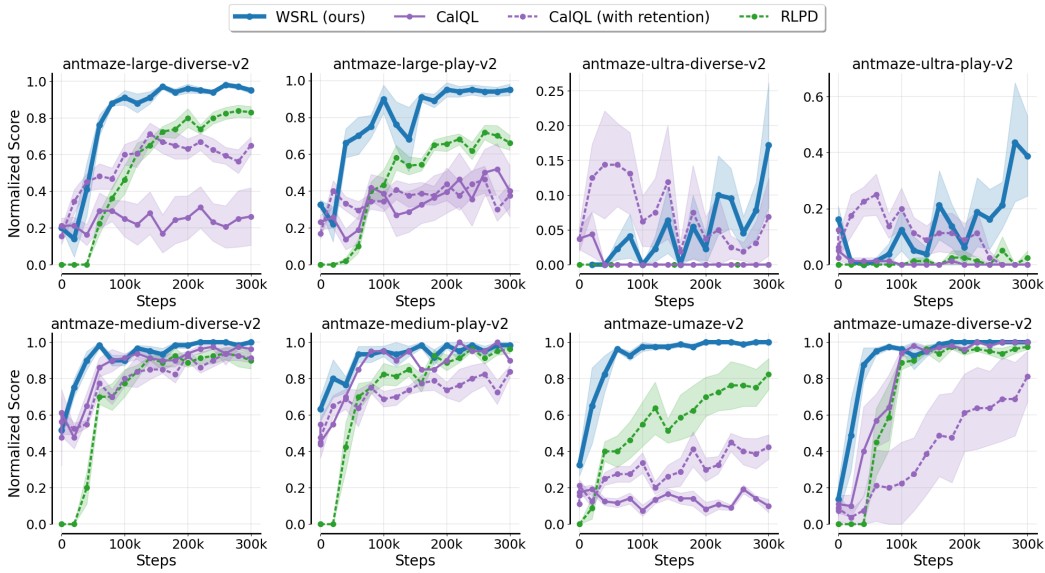

Figure 14: WSRL on all eight D4RL antmaze environments, along with RLPD and CalQL baselines. Step 0 shows the start of fine-tuning for WSRL and CalQL, and start of RLPD. Solid lines do not retain offline data, while dotted lines do.

## B   RESULTS ON MUJOCO LOCOMOTION ENVIRONMENTS

Additionally, we also apply WSRL on nine different Mujoco locomotion domains in the no-retention fine-tuning setting. Specifically, we experiment with three different robot embodiments (`Halfcheetah`, `Hopper`, and `Walker`), each with three different types of datasets. The `random` datasets are collected with a random policy; the `expert` datasets are collected with a policy trained to completion with SAC; and the `medium-replay` datasets are collected with the replay buffer of a policy trained to the performance approximately $1/3$ of the expert. As Figure 15 shows, WSRL outperforms or is similar to the best baseline methods.

For WSRL, the hyperparameters were exactly as those in Section 5 and listed in Appendix I with one exception: its pre-trained policy and value function are done with CQL offline training instead of CalQL. This is because these offline datasets have dense rewards and do not end in a terminal state, and therefore do not have ground-truth return-to-go to support the CalQL regularizer. For the same reason we did not include a CalQL baseline in Figure 15. Both the IQL and CQL baseline in Figure 15 do not retain offline data, and use an ensemble of 10 Q functions, along with layer normalization in the Q functions. RLPD does retain offline data.

## C   EXPERIMENTAL SETUP

**(1)** The `Antmaze` tasks from D4RL (Fu et al., 2020a) are a class of long-horizon navigation tasks that require controlling an 8-DOF Ant robot to reach a goal with a sparse reward. The agent has to

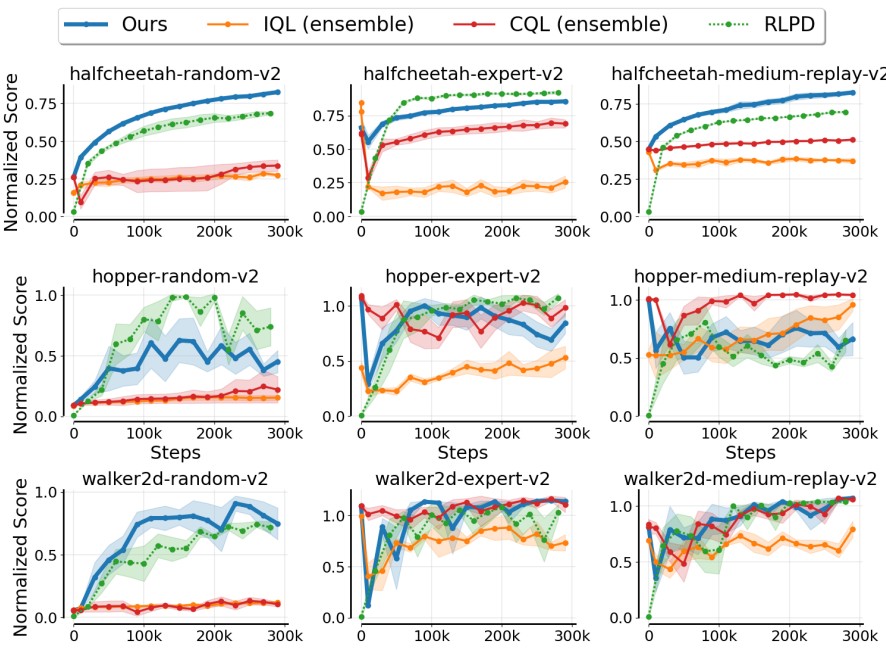

Figure 15: WSRL on nine Mujoco locomotion environments with dense rewards, along baselines. Step 0 shows the start of fine-tuning for WSRL and CalQL, and start of RLPD. Solid lines do not retain offline data, while dotted lines do.

learn to "stitch" experiences together from a suboptimal dataset. In addition to the original mazes from D4RL, we include `Antmaze-Ultra` (Jiang et al., 2022), a larger and more challenging maze. We only include three of the hardest Antmaze environments in Section 5, and provide results on all eight Antmazes in Appendix A. **(2)** The `Kitchen` environment is a long-horizon manipulation task to control a 9-DoF Franka robot arm to perfrom 4 sequential subtasks in a simulated kitchen. **(3)** The `Adroit` environments are a suite of dexterous manipulation tasks to control a 28-DoF five-fingered hand to manipulate a pen to desired position, open a door, and relocating a ball to desired position. The agent observes a binary reward when it succeeds. Each data has an offline dataset that provides a narrow offline dataset of 25 human demonstrations and additional trajectories collected by a behavior cloning policy. **(4)** The `Mujoco Locomotion` environments in D4RL are dense reward settings where agents learn to control robotic joints to perform various locomotion tasks.

## D  ABLATION STUDIES ON WARMUP PHASE

**Impact of different warmup types.** One natural question arises: why does the simple approach of warming up the replay buffer significantly boost performance during fine-tuning? One hypothesis is that seeding the replay buffer with some data helps prevent early overfitting, as much work has found in online RL (Nikishin et al., 2022). To test this hypothesis, we plot in Figure 16 in the fine-tuning performance of initializing with random actions, and compare it to initializing with pre-trained policy actions as well as not initializing the buffer at all. It is clear that seeding the buffer with random actions significantly underperform the warmup approach, and in fact does not even provide much benefit as compared to not seeding the buffer at all. This suggests that the reason warmup phase helps is not because of preventing overfitting.

**Impact of different length warmups.** Since warmup phase seems to be critical to efficient online fine-tuning, we study whether the length of this warmup phase impacts fine-tuning performance. Figure 17 shows warmup phase of lengths 1k, 5k, and 20k on three different environments. It is clear that short warmup phase (1k) sometimes lead to worse asymptotic performance or instability during fine-tuning. On the other hand, longer warmup phases could also hurt (e.g. on **Kitchen-mixed**) because it adds too much offline-like data into the replay buffer and slows down online

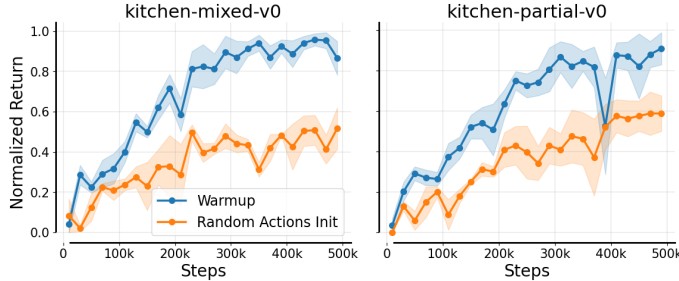

Figure 16: Comparing seeding the buffer with random actions to actions from the pre-trained policy: initializing with the pre-trained policy action works significantly better on `kitchen-mixed` (left) and `kitchen-partial` (right).

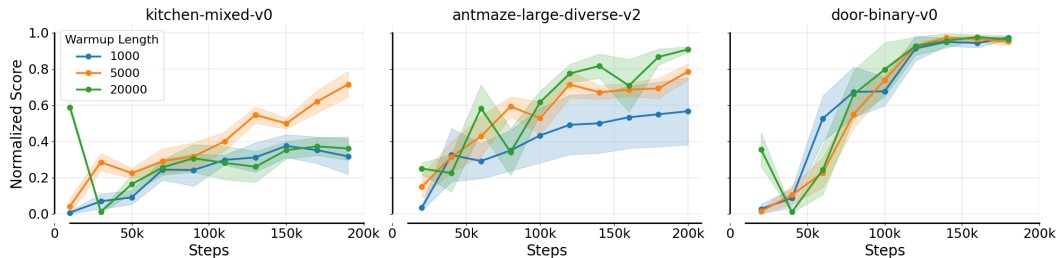

Figure 17: Impact of warmup phase of length 1k, 5k, 20k on `Kitchen-mixed` (left), `Antmaze-large-diverse` (middle), and `Door-binary` (right).

improvement. In WSRL, we did not tune the lengths of the warmup phase beyond what is shown in Figure 17, and we use 5000 warmup steps for all environments.

# E UNLEARNING AND RECOVERY AT THE START OF FINE-TUNING: AN ANALYSIS

To further illustrate the behavior of offline-to-online RL agents at the start of online fine-tuning, we show in Figure 18 the performance of WSRL, along with two other algorithms, evaluated at much smaller intervals than Figure 7. We fine-tune all agents for $50,000$ steps online across six different environments, and evaluate every $2,000$ environment steps.

Figure 18 shows that WSRL experiences an initial dip in policy performance after $5,000$ steps of warmup, but recovers much faster than CQL and CalQL. We hypothesize that such a dip might be inevitable at the start of online fine-tuning in the no offline data retention setting because the policy is experiencing different states than what it was trained on and potentially states it has never seen (We analyze this with much more detail below). Moreover in some environments (e.g., binary reward environments), one might expect small fluctuations in the policy to manifest as large changes in the actual policy performance. However, such brief performance dip does not mean the policy/Q function has been catastrophically destroyed, which is evidenced by the fact that WSRL recovers faster than its peer algorithms and learns faster than online RL algorithms such as RLPD (See Figure 9). If this initial dip would have destroyed all pre-training knowledge from the policy, then we would not expect quick recovery.

In fact, in general, it is impossible to build a no data retention fine-tuning algorithm whose performance does not initially degrade as we move from offline data to online training on all environments and offline data compositions (Xie et al., 2021). Intuitively, this is because it violates a sort of "no free lunch" result: for example, consider a sparse reward problem where the reward function is an arbitrary non-smooth function over actions, here even a minor change in policy action results in a catastrophic change in return. Therefore just deducing whether an algorithm has lost is prior or not based on performance may not be the most informative. Instead, a more meaningful metric to mea-

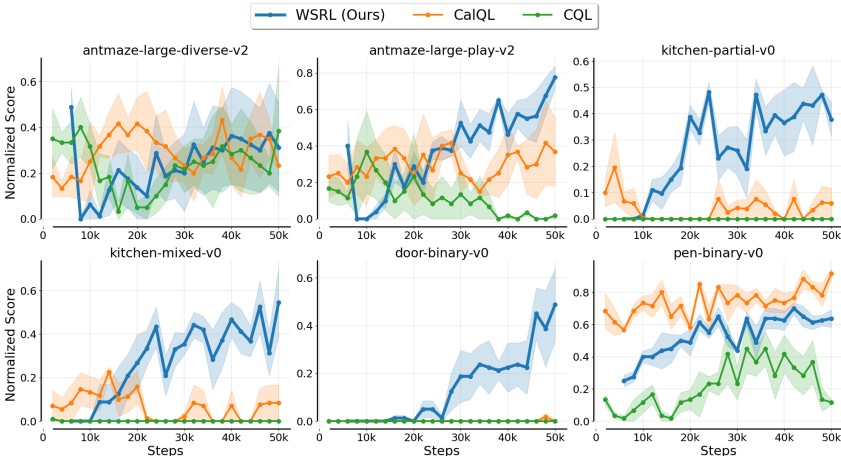

Figure 18: First 50,000 steps of fine-tuning with denser evaluation intervals. Step 0 in the plot show the start of online fine-tuning. WSRL starts being evaluated after $K = 5000$ steps of warmup. All agents are evaluated in the no-retention fine-tuning setting.

sure catastrophic forgetting is to evaluate how much a fine-tuning algorithm with no data retention deviates from its pre-training, and how fast it can adjust to the online state-action distribution.

Therefore, to investigate how much the policy and Q-function have deviated from its offline-pretraining during the initial "performance dip", we plot the KL divergence between the pre-trained policy/Q-function and the fine-tuned ones. Figure 19 shows the KL divergence between the policies $D_{\mathrm{KL}}(\pi_\psi^{\mathrm{pre}}||\pi_\psi)$ evaluated on both the online distribution and the offline distribution. In Figure 19 (Top), we can see that $D_{\mathrm{KL}}(\pi_\psi^{\mathrm{pre}}||\pi_\psi)$ on the offline distribution generally increases during fine-tuning for all three agents. This increase indicates that the fine-tuned policy has deviated from the pre-trained policy on at least some parts of the dataset distribution. This is actually expected in the no-retention fine-tuning setting because of the distribution shift from offline to online. To be more specific, for example, in Antmaze environments, the offline dataset exhibits a very diverse state-action distribution, covering almost all the locations in the entire maze, while fine-tuning is a single-goal navigation task. In no-retention fine-tuning, the agent is incentivized to forget about parts of the offline dataset that is irrelevant to the fine-tuning task and specialize to the online task. Compared to CQL and CalQL, WSRL generally has the same asymptotic value for $D_{\mathrm{KL}}$ but reaches convergence much faster. This suggests that WSRL actively adapts to the online distribution much quicker compared to its no-retention counterparts, perhaps thanks to its non-conservative objective optimized solely on online data and its high update-to-data ratio during online RL.

On the other hand, Figure 19 (Bottom) shows $D_{\mathrm{KL}}$ on the online distribution for WSRL increases slightly, but is much smaller compared to CQL and CalQL (without data retention). This indicates that WSRL's policy remains almost the same on the online distribution. This is desirable because the pre-trained policy already has decent performance, and a capable fine-tuning algorithm should not forget that capability while adjusting slightly to unseen (but not out-of-distribution) states. In summary, due to the distribution shift from offline pre-training to no-retention fine-tuning, WSRL is forgetting experience in the offline dataset that was learned during offline pre-training but is in reality irrelevant for specializing to the online task. In contrast, it is instead specializing to the online task.

In addition to the above analysis on the policy, in Figure 20, we plot the KL divergence of pre-trained Q-function $Q_\theta^{\mathrm{pre}}$ and fine-tuned Q-function $Q_\theta$. We normalize the Q-values into a distribution with softmax and plot $D_{KL}(softmax(Q_\theta^{\mathrm{pre}})||softmax(Q_\theta))$. We evaluate the $D_{KL}$ on states sampled from both the offline dataset distribution (Figure 20 Top) and the online replay buffer distribution (Figure 20 Bottom), and on actions sampled from a equal-weighted mix of $\pi_\psi^{\mathrm{pre}}(s)$ and $\pi_\psi(s)$. The results show that CQL and CalQL Q-functions usually diverge significantly from pre-training, probably due to the downward spiral phenomenon described in Section 3. In comparison, WSRL's

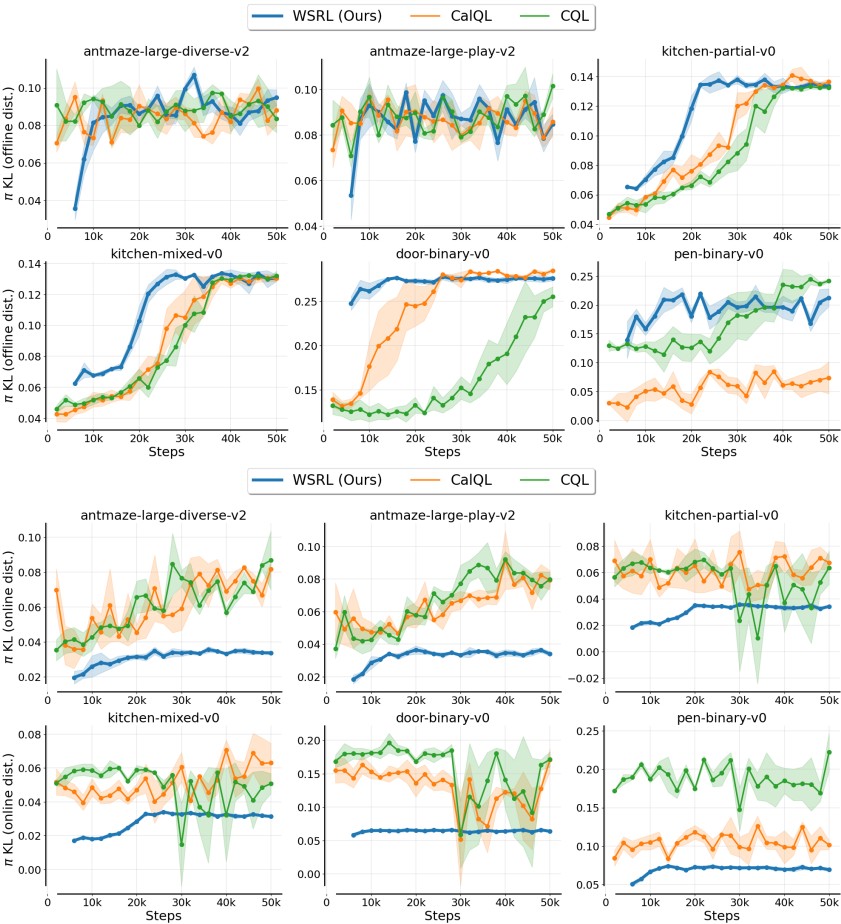

Figure 19: **Policy KL divergence** $D_{KL}(\pi_\psi^{\mathrm{pre}}||\pi_\psi)$ between the pre-trained offline policy and the fine-tuned online policy at the first $50k$ steps of fine-tuning. The top plot shows the KL divergence evaluated on the offline dataset distribution; the bottom plot shows the KL divergence on the online state-and-action distribution, sampled from the replay buffer. WSRL is not plotted during the first $5,000$ steps of warmup. CQL and CalQL do not retain offline data.

Q-function remains stable on both the online and offline distributions, suggesting WSRL did not forget its pre-trained Q-function.

In summary, the above analysis on the KL divergence suggests that WSRL remains stable and **quickly adapts** during fine-tuning and **does not forget priors learned from offline pre-training**.

## F  DOES FREEZING THE POLICY AT THE START OF FINE-TUNING HELP?

Since Appendix E shows a brief period at the start of fine-tuning where policy performance take a dip, one natural question is whether such a dip is avoidable. The most straight forward way to avoid such a drop in policy performance is to freeze the policy during initial fine-tuning. In other words, for $N$ steps at the on set of fine-tuning, we only pass the gradients through the Q function and train the Q function, but freeze the policy. After $N$ online steps, we start training both the policy and the Q function.

Figure 21 shows the performance of WSRL after freezing the policy for $N = \{10k, 30k\}$ steps. It's obvious that even when we freeze the policy for some number of steps to let the Q-function adjust online, the policy still suffers a dip after it is unfrozen. In fact, this is somewhat an expected result because the policy needs to adjust to the OOD online state-action distribution, as well as the new online Q-function, and such adjustment process is expected to make the policy performance worse.

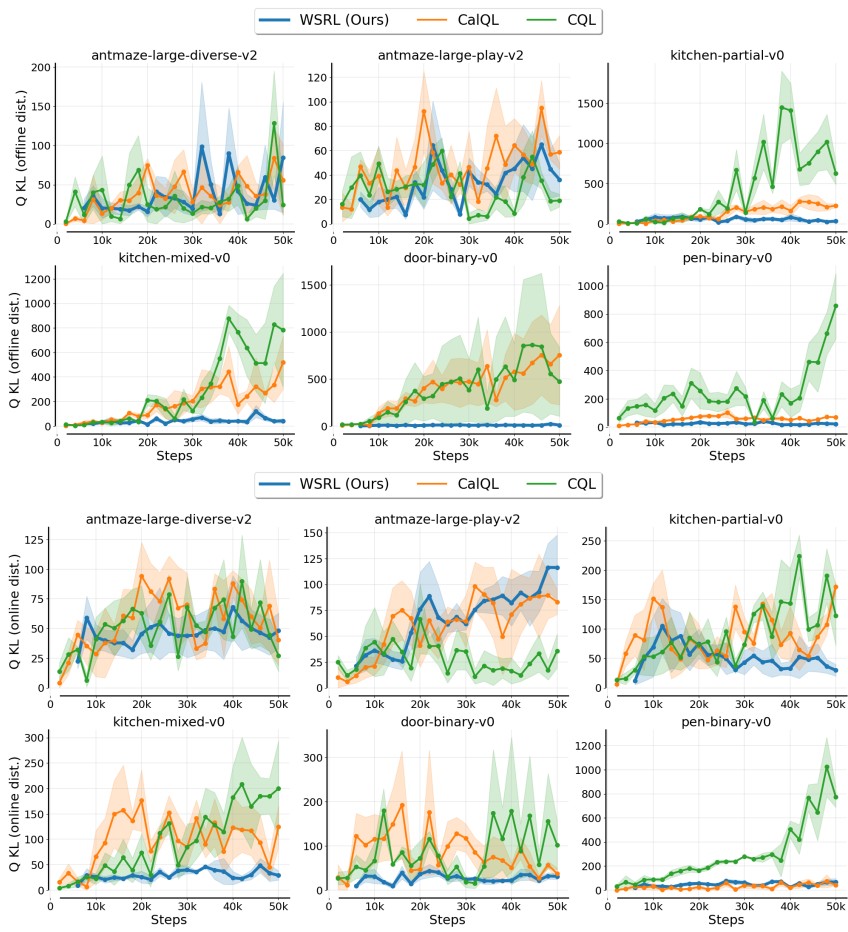

Figure 20: **Q-function KL divergence** $D_{KL}(softmax(Q_\theta^{\mathrm{pre}})||softmax(Q_\theta))$ between the pre-trained Q-function and the fine-tuned Q-function at the first $50k$ steps of fine-tuning. We evaluate the Q-functions by sampling states from the offline dataset distribution (Top) and the online buffer distribution (Bottom), and we sample actions by running $\pi_\psi^{\mathrm{pre}}$ and $\pi_\psi$ on the sampled states. WSRL is not plotted during the first $5,000$ steps of warmup. CQL and CalQL do not retain offline data.

## G   WHY WARM-UP PREVENTS Q-VALUES DIVERGENCE

In WSRL, the policy and value function is pre-trained offline with CalQL (Nakamoto et al., 2024), and the online fine-tuning process is done with SAC (Haarnoja et al., 2018a). This change of RL algorithm could lead to miscalibration issues, where the pre-trained values are more pessimistic than ground truth values. As we have shown in Section 3, this hurts fine-tuning when it backs up a pessimistic target Q-value through the Bellman update. This particularly hurts when the Bellman target is computed on an OOD state-action pair, because OOD state-action pair have more pessimistic values than state-action paris seen in the offline dataset, by the nature of pessimistic pre-training. If there were no warm-up phase, the agent will collect OOD data into the buffer, leading to Bellman backups with pessimistic target Q values, which in turn leads to Q-divergence. This is the "downward spiral" phenomenon in Section 3. However, warmup solves this problem by putting more offline-like data into the replay buffer where Q-values are not as pessimistic, thereby preventing the downward spiral in the online Bellman backups and uses high UTD in online RL to quickly re-calibrate the Q-values.

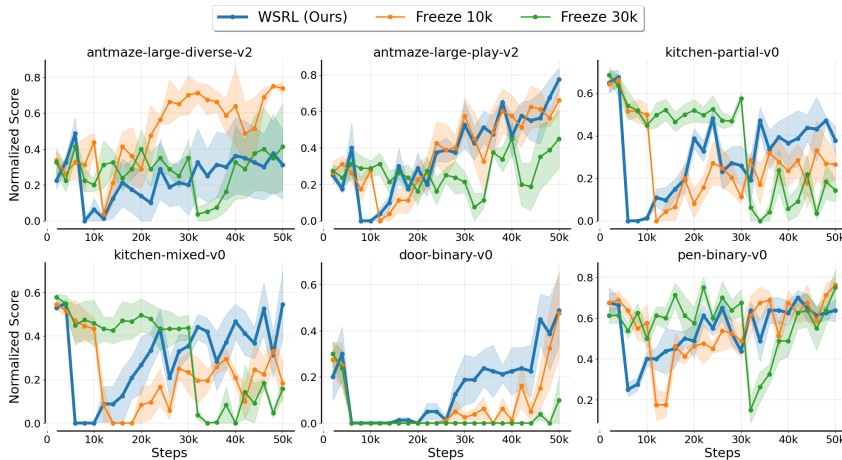

Figure 21: Freezing the policy for $N \in \{10k, 30k\}$ steps at the onset of fine-tuning doesn't prevent the performance dip. In the plot, we show policy performance of WSRL vs. WSRL with initial policy freeze across six environments. Step 0 is the start of online fine-tuning.

## H  ABLATION STUDIES ON DIFFERENT TYPES OF VALUE INITIALIZATION

**WSRL is agnostic to the offline RL pre-training algorithm.** Furthermore, we find that it is not crucial which specific offline RL algorithm we use to obtain the pre-trained Q values. In Figure 22, we show that the Q-values from IQL, CQL, and CalQL work just as well on three different environments, even though CQL optimizes for conservative Q-values, CalQL is less conservative, and IQL is not conservative at all. In particular, we observe that Calibrated Q-values as an initialization provides some small performance benefits on `Kitchen-mixed`. Therefore, we use calibrated Q-values from CalQL offline pre-training for our main experiments.

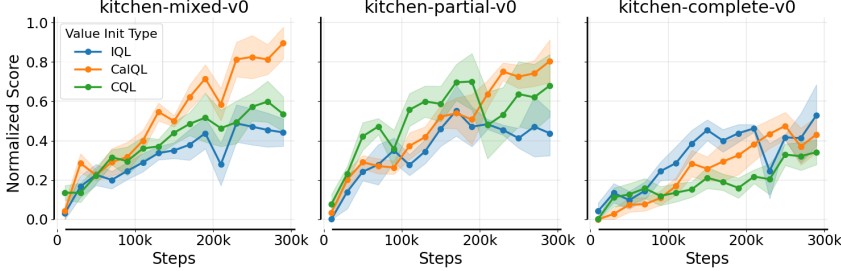

Figure 22: The offline RL algorithm used to pre-train the Q-values does not affect performance: for `Kitchen-mixed` (left), `Kitchen-partial` (middle), and `Kitchen-complete`, WSRL is able to achieve similar performances by initializing with pre-trained values from CQL, IQL, and CalQL, though CalQL initializations have small benefits on `Kitchen-partial`.

## I  IMPLEMENTATION DETAILS

Code for WSRL is released at https://github.com/zhouzypaul/wsrl.

**Pseudocode.** See Algorithm 1. During online RL updates, the critic is updated with standard temporal difference loss and the actor is updated with policy gradient (in this case, a reparameterization based policy gradient estimator) with entropy regularization as in Soft Actor Critic (Haarnoja et al., 2018a).

**WSRL Hyperparameters.** We use 5K warmup steps ($K = 5,000$). For the online RL algorithm in WSRL, we use the online SAC (Haarnoja et al., 2018b) implementation in RLPD (Ball et al., 2023) with a UTD of 4 and actor delay of 4 (update the actor once for every four critic steps), batch size

---

**Algorithm 1** WSRL: **W**arm **S**tart **R**einforcement **L**earning

---

**Require:** Offline RL algorithm $\mathcal{A}_{\text{off}}$, Pre-training dataset $\mathcal{D}_{\text{off}}$.

$\qquad Q_\theta^{\text{pre}}, \pi_\psi^{\text{pre}} \leftarrow \text{TrainOffline}(\mathcal{A}_{\text{off}}, \mathcal{D}_{\text{off}})$ $\qquad\qquad\qquad\qquad$ ▷ Offline RL pre-training

**Require:** $Q_\theta^{\text{pre}}, \pi_\psi^{\text{pre}}$, Online RL algorithm $\mathcal{A}_{\text{on}}$ with UTD $M$, Replay buffer $\mathcal{R} \leftarrow \emptyset$, warmup step

$\quad K$.

$\qquad Q_\theta \leftarrow Q_\theta^{\text{pre}}, \pi_\psi \leftarrow \pi_\psi^{\text{pre}}, \mathcal{R} \leftarrow \emptyset$ $\qquad\qquad\qquad\qquad\qquad\qquad$ ▷ Initialization

$\qquad$ **while** step $\leq$ max steps **do**

$\qquad\qquad$ **if** step $\leq K$ **then**

$\qquad\qquad\qquad (s, a, s', r) \leftarrow \text{interact}(\pi_\psi^{\text{pre}}, \text{environment})$ $\qquad\qquad\qquad$ ▷ Warmup Phase

$\qquad\qquad$ **else**

$\qquad\qquad\qquad (s, a, s', r) \leftarrow \text{interact}(\pi_\psi, \text{environment})$

$\qquad\qquad$ **end if**

$\qquad\qquad \mathcal{R} \leftarrow \mathcal{R} \cup \{(s, a, s', r)\}$

$\qquad\qquad$ **if** step $> K$ **then**

$\qquad\qquad\qquad \text{Batch}_1, \text{Batch}_2, ..., \text{Batch}_M \sim \mathcal{R}$

$\qquad\qquad\qquad Q_\theta \leftarrow \text{TemporalDifferenceUpdate}(Q_\theta, \text{Batch}_i)$ for $M$ times $\qquad$ ▷ High UTD Critic Update

$\qquad\qquad\qquad \pi_\psi \leftarrow \text{PolicyGradient}(\pi_\psi, \text{Batch}_1 \cup \cdots \cup \text{Batch}_M)$ ▷ Actor Update with Actor Delay of $M$

$\qquad\qquad$ **end if**

$\qquad$ **end while**

---

of 256, actor learning rate of $1e-4$, critic learning rate of $3e-4$, and temperature learning rate of $1e-4$. We use and ensemble of 10 Q functions, and predict the Q value by randomly sub-sampling 2 and taking the min over the 2 Q-functions (Chen et al., 2021). When we initialize the policy network and the Q-function network from offline RL pre-training, we keep the optimizer state of these networks.

In antmaze environments, we find it important to calculate the TD-target by taking the maximum of the TD-target over the ensemble of 10 Q-functions as done by Kumar et al. (2020). This design does not affect performance on the other environments. We hypothesize that this is because antmaze environments require more optimism online, cooperating the design decisions by Ball et al. (2023) as described below.

**Environment Details.** All environments use a discount factor of 0.99. We set the reward scale in the benchmark tasks following previous work (Nakamoto et al., 2024; Kostrikov et al., 2021; Kumar et al., 2020). In Antmaze and Adroit environments, we use sparse reward of 5 at the goal and $-5$ at each step. In Kitchen, we use 0 at the goal, and $-1$ for each subtask that is not completed at the current timestep, giving possible reward values $-4, -3, -2, -1, 0$ [2]. In Mujoco locomotion environment, we use the original environment dense rewards. We pre-train 1M steps on Antmaze, 20k steps on Adroit, 250k steps on Kitchen and Mujoco locomotion.

**Baseline Hyperparameters.** All SAC-based methods use the same learning rate as WSRL above, and IQL-based methods use actor and critic learning rate of $3e-4$. All methods that have high update-to-data ratio or an ensemble of Q-functions use layer normalization as a regularization. The method-specific details are listed below, with hyperparameters gotten from the original papers with no further tuning.

$\quad$ **IQL.** Antmaze environments use expectile 0.9, and all other environments use expectile 0.7. For the inverse temperature in the actor, we use 10 in Antmaze, 3 in Mujoco locomotion, and 0.5 for Kitchen and Adroit. Unless otherwise specified, IQL uses two Q functions and takes the minimum over them to estimate the Q value.

$\quad$ **CQL / CalQL.** In Antmazes, we use the dual version of the CQL objective and set gap to 0.8. In Mujoco locomotion and Kitchen environments, we set CQL regularizer weight $\alpha = 5$.

---

[2] In the main performance comparison results in Figures 7 and 9, kitchen environments use a maximum episode length of 280 as the default from D4RL, while other ablation and analysis experiments may use a version of kitchen by Nakamoto et al. (2024) that has maximum episode length of 1000 but no other changes. All comparisons plotted are on the same kitchen version.

In Adroit environments, we set $\alpha = 1$. CQL and CalQL use the same CQL regularizer for both pre-trainign and fine-tuning. In experiments that do retain offline data, each update batch samples $50\%$ from the offline data on Antmaze, Adroit, and Mujoco locomotion environments, and $25\%$ on Kitchen environments (Nakamoto et al., 2024). Unless otherwise specified, CQL/CalQL uses two Q functions and takes the minimum over them to estimate the Q-value.

**RLPD / SAC(fast).** We use an ensemble of $10$ Q-functions and a UTD of $4$ with batch size $256$. Following (Ball et al., 2023), in antmaze environments, we predict the Q-value from the ensemble by randomly subsampling $1$ Q-function. This is needed for more optimism online in antmaze environments. In all other environments, we subsample $2$ Q-functions and take the minimum over them to estimate the Q-value.

**JSRL.** Same hyperparameters as RLPD, where we improve JSRL's competitiveness with a Q-ensemble and UTD. For each online interaction episode, JSRL decides whether to roll in the pre-trained frozen policy or roll out the fine-tuned policy with probability $0.5$. This probability decreases linearly to $0$ over the first $100k$ fine-tuning steps. In episode where the JSRL decides the roll in the pre-trained frozen policy, the number of step it rolls in follows a geometric distribution with $\gamma = 0.99$, after which the fine-tuned policy is rolled out until the end of the episode.

**SO2.** Same hyperparameters as RLPD. Different from the original paper, we use $10$ Q-ensembles to make a fair comparison with WSRL and other baselines which have the same number of Q-ensembles. For the action noise, we use a standard normal with variance $0.3$, and we clip the action noise to be between $(-0.6, 0.6)$.

## J  WARMUP WITH TRANSITIONS FROM THE OFFLINE DATASET

We have shown in Section 5 that the warmup period is essential for efficient fine-tuning with WSRL. One interesting question is whether such warmup data can be collected by sampling the offline dataset, instead of online interactions with the frozen pre-trained policy as in WSRL. Therefore, we run an ablation experiment in Figure 23 where we replace the $5,000$ steps of warmup period by initializing the online replay buffer with $5,000$ random transitions sampled from the offline dataset, which we will refer to as "Dataset Warmup". As Figure 23 shows, while the two methods are similar on `Adroit` environments, WSRL is slightly better in `Kitchen` and much better on `Antmaze`. This is perhaps because the 5000 randomly sampled transitions might not be relevant to the online fine-tuning task, especially in `Antmaze` where the dataset has diverse state-action coverage (See Appendix E for a more detailed discussion). When the replay buffer is initialized with less relevant data, it is less effective at preventing Q-value divergence (Section 3) and recalibrating the online Q-function and policy. This perhaps highlights the utility of our approach: despite not having access to *any* offline data, WSRL is able to achieve similar or better performance than using transitions from the offline dataset in the no-retention fine-tuning setting.

## K  WSRL WITH OFFLINE DATA RETENTION

In the main paper, we have shown that WSRL can efficiently fine-tune without retaining the offline pre-training dataset. One natural question arises: can WSRL do even better if we allow offline data retention? To answer this question, we run WSRL with the online replay buffer initialized with the whole offline dataset. Figure 24 shows that on average, retaining the offline data does not give WSRL any advantages, probably because it already has the necessary knowledge in the offline policy and Q-function; WSRL is actually a bit faster later on in fine-tuning, perhaps due to the fact that it is updating the policy on more online data.

## L  ABLATING THE EFFECTS OF Q-ENSEMBLE AND LAYER NORMALIZATION

In WSRL, we choose to use the most effective online RL algorithm fine-tune with high update-to-data (UTD) ratio. Following the design choices by Ball et al. (2023), we also use an ensemble of 10 Q-functions and layer normalization as regularization to stabilize online training in the high UTD regime. In Figure 7, SAC (fast) and JSRL also has high UTD and therefore we also apply both regularizations. However, we implement IQL, CQL, CalQL without these regularizations, as in the

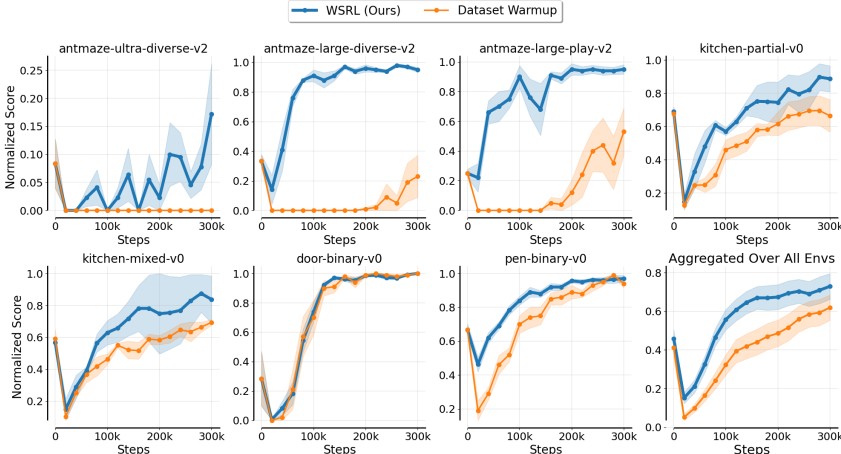

Figure 23: **Warming up with transitions from the offline dataset** is less effective than warming up with online interactions (WSRL).

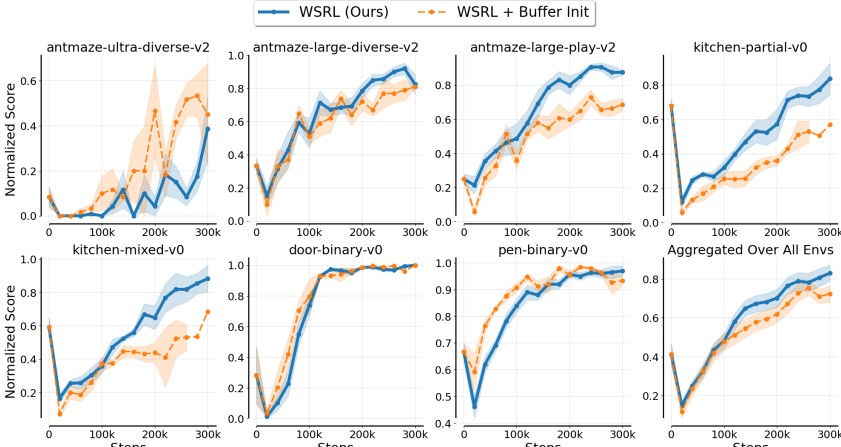

Figure 24: **Initializing the replay buffer with the offline dataset** does not give WSRL any advantage during fine-tuning, and may make it a bit slower.

original papers. To ablate the effects of layer normalization and Q-ensemble in no-retention fine-tuning, we apply both to each of IQL, CQL, and CalQL. In Figure 25, we apply layer normalization after each dense layer in the actor and the critic MLP, and find that it minimally affect performance. In Figure 26, we apply layer normalization as well as a Q-ensemble, and refer to the combination as REDQ (Chen et al., 2021). We find that REDQ helps significantly on Antmaze tasks, but not huge gains in other environments. Overall, WSRL still significantly outperforms IQL, CQL, and CalQL with extra regularizations.

# M  WSRL WITH DIFFERENT UPDATE-TO-DATA RATIOS

In Section 5, we mainly experiment with UTD=4 for all methods. One interesting question is whether WSRL can benefit from an even higher UTD, and how it compares with other fast online RL methods with higher UTDs. In Figure 27, we compare WSRL against RLPD under UTD 20, and find that while UTD 20 improves performance slightly, the difference is not huge.

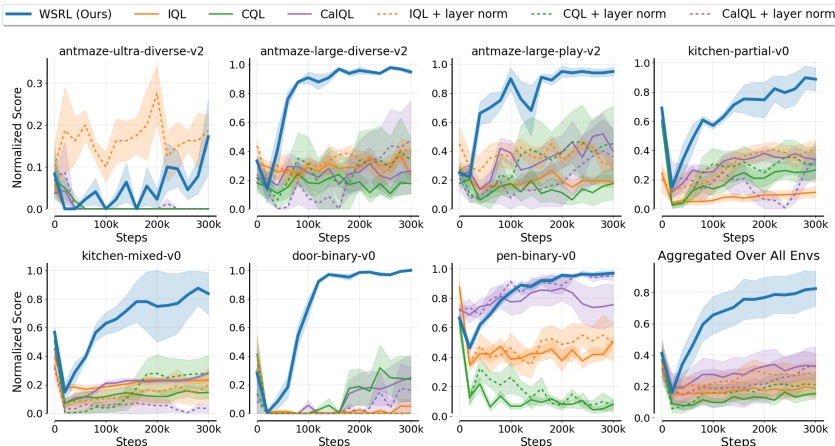

Figure 25: **Layer normalization** does not impact the performance of IQL, CQL, CalQL in no-retention fine-tuning.

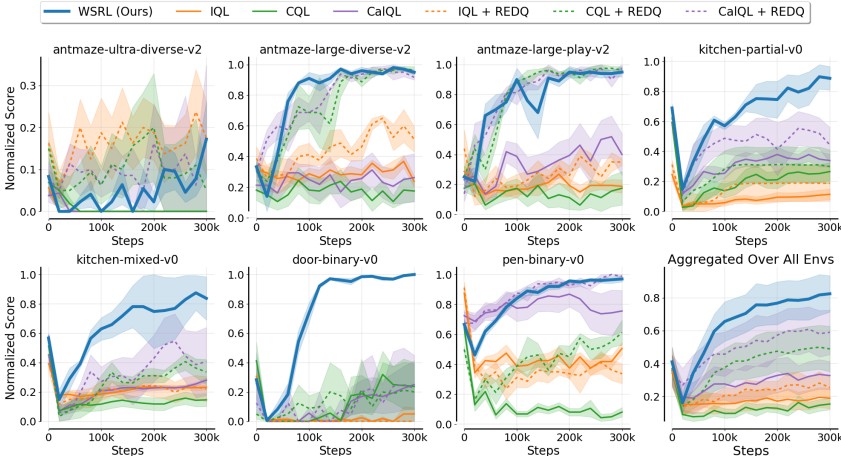

Figure 26: Impact of **layer normalization and Q-ensemble** on IQL, CQL, CalQL in no-retention fine-tuning: it benefits `Antmaze` environments greatly, but not so much on other environments.

## N  WSRL WITH VARYING LEVELS OF OFFLINE POLICY

We investigate how WSRL performs with varying levels of expertise of the offline pre-trained policy. Specifically, we consider `Kitchen-complete-v0` and `Relocate-binary-v0`, two especially hard tasks for offline RL where pre-training with CalQL leads to poor performance. In `Recolate-binary-v0`, CalQL completely fails and has pre-trained performance near 0; CQL and IQL also has pre-training performance 0, indicating that this task is inherently hard for offline RL agents. In `Kitchen-complete-v0`, CalQL (15.47%) significantly underperforms IQL (70.83%) despite our tunning efforts, which suggests there is some inherent limitation in CalQL learning a good Q-funciton in this domain. Not surprisingly, Figure 28 shows that WSRL also performs poorly: while WSRL can learn somewhat in `Kitchen-complete-v0` with a non-zero initialization, it completely fails to learn in `Adroit-binary-v0`. This is expected because when pre-training fails, initializing with the pre-trained network may not bring any useful information gain, and may actually hurt fine-tuning by reducing the network's plasticity (Nikishin et al., 2022), a known issue in online RL. In such situations, one may wish to resort to different methods (e.g. online RL) rather than fine-tuning from a bad initialization.

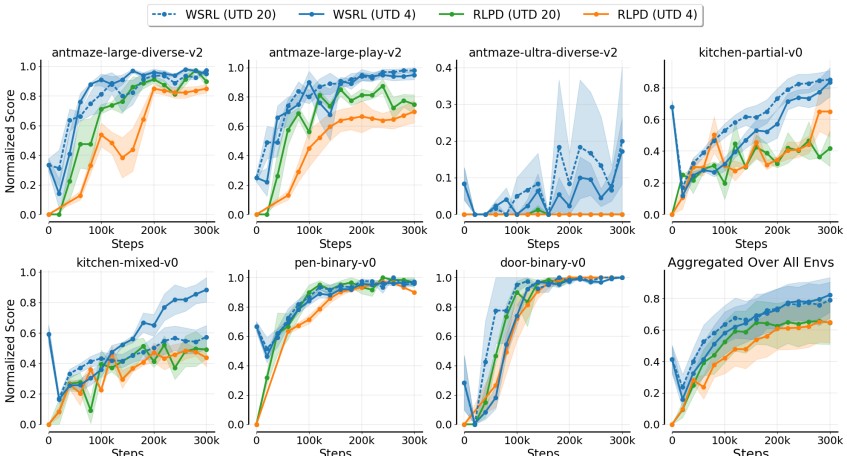

Figure 27: **UTD 20 vs. UTD 4:** For both WSRL and RLPD, UTD 20 performs only slightly better than UTD 4.

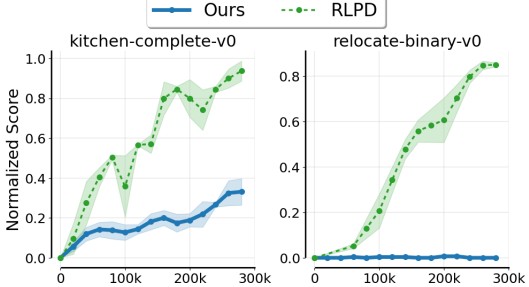

Figure 28: On environments where the pre-training completely fails, WSRL does not work well.

