# OpenReview forum: "Efficient Online Reinforcement Learning Fine-Tuning Need Not Retain Offline Data"
_ICLR.cc/2025/Conference — ICLR 2025 Poster_

### Official Review · Reviewer_7N5M · 2024-10-30

**Soundness:** 3
**Presentation:** 3
**Contribution:** 2
**Rating:** 8
**Confidence:** 3

**Summary:**

In this paper, the authors highlight the requirement for online fine-tuning algorithms to retain offline data in their replay buffers to remain effective. The authors thus aim to remove this requirement, by instead using a small amount of online data collected from rollouts of the frozen pre-trained policy. They show that the TD-error for the online policy under the offline data distribution diverges as less offline data is retained. The postulate that this is due to offline/online data distribution mismatch.

**Strengths:**

* There is a significant decrease in the efficacy of many finetuning algorithms when offline data is not retained, and this appears to be the first work to focus on this issue
* The experiments supporting the hypothesis of Q-value divergence over the offline data distribution are convincing
* The results are proven over many environments
* Clear layout and explanations

**Weaknesses:**

* No investigation of algorithm performance over varying levels of expertise of the offline pre-trained policy - it would have been interesting to see WSRL results when initialized from pre-trained policies of different quality levels or trained for varying numbers of steps.
* A comparison of RLPD initialised with prior policy, but without the additional online data should have been included as a baseline.

Minor
Line 84: an warmup

Line 207: on which we measure metrics on

Line 253: onto the previous action pair that the TD error (?)

Figure 4: Mixed, complete, partial (what does this refer to?)

**Questions:**

1. In Figure 7, JSRL is shown as not retaining offline data. However, JSRL is implemented over IQL in Uchendu et al., and the IQL implementation linked in the paper does retain offline data. Did the authors specify to you that they chose not to retain offline data?
2. It is hypothesised in section 4.1 that backups of small, over-pessimistic Q-values from pessimistic offline algorithms cause the Q-values divergence. However, in Figure 9 you mention the warm-up helps avoid over-pessimistic values for WSRL. I had understood that WSRL used an online, non-pessimistic algorithm (RLPD) as its base, so what would cause the underestimation in this case? Would RLPD work if initialised to the policy learned offline, without adding any data collected online?

---

> ### Author Response · Authors · 2024-11-20
> **Paper updates and clarifications**
>
> Thank you for your constructive and encouraging feedback! In response to your comments, we 1) answer below your question on why online RL algorithms still suffers from Q-value underestimation issues, 2) added Figures showing WSRL with different levels of expertise in the offline policy, 3) point you to our baseline comparison on RLPD-initialized with prior policy, and 4) anwer your question of why JSRL does not retain offline data. In addition, we also added extensive experiments on nine Gym locomotion tasks and eight antmaze environments, as well as many other ablations. Please let us know if you have any other questions.
>
> > It is hypothesized in section 4.1 that backups of small, over-pessimistic Q-values from pessimistic offline algorithms cause the Q-values divergence. However, in Figure 9 you mention the warm-up helps avoid over-pessimistic values for WSRL. I had understood that WSRL used an online, non-pessimistic algorithm (RLPD) as its base, so what would cause the underestimation in this case?
>
> Good question! As you mentioned, Section 4.1 explains how over-pessimistic Bellman backup leads to Q-values divergence. You’re right that WSRL uses an online non-pessimistic algorithm (RLPD), but it initializes its Q function from the pessimistic offline pre-training, e.g., with Cal-QL or CQL. By the nature of pessimistic pre-training, OOD state-action values may be especially smaller in value compared to the state-action pairs seen in the offline dataset. During fine-tuning, if there were no warm-up, the algorithm will collect OOD data into the buffer, and so Bellman backups with pessimistic target Q values will lead to Q-divergence. However, warmup solves this problem by putting more offline-like data into the replay buffer where Q-values are not as pessimistic, thereby preventing the pathologies in Section 4.1.
>
> > It would have been interesting to see WSRL results when initialized from pre-trained policies of different quality levels or trained for varying numbers of steps.
>
> Following your suggestion, we added Appendix J to show WSRL on varying levels of expertise on the offline policy. Specifically, Figure 21 shows WSRL performance on kitchen-complete-v0 and relocate-binary-v0, two especially hard tasks for offline RL where CalQL has pre-trained performance 0. Not surprisingly, WSRL is also not able to learn: when pre-training fails, indicating that offline training is not able to learn useful behavior, initializing with the pre-trained networks may not bring any information gain from pre-training, and may actually hurt fine-tuning by reducing the network’s plasticity [1].
>
> > A comparison of RLPD initialized with prior policy, but without the additional online data should have been included as a baseline.
>
> This is actually included in Figure 11 as an ablation; When WSRL is not initialized with the Q-function, it is equivalent to RLPD with no offline data retention (Note that RLPD is initialized with random actions, but policy-initialized-RLPD is equivalent to doing warmup). Results show that WSRL outperforms this baseline. Specifically, we showed that WSRL is much better than policy-initialized-RLPD on Antmaze environments, moderately better on Adroit, and similar on Kitchen.
>
> > In Figure 7, JSRL is shown as not retaining offline data. However, JSRL is implemented over IQL in Uchendu et al., and the IQL implementation linked in the paper does retain offline data. Did the authors specify to you that they chose not to retain offline data?
>
> You are correct, IQL usually retains the offline data during offline-to-online fine-tuning. However, In the Uchendu el al.[2], Algorithm 1 (page 4) bullet point #2 says that buffer D is initialized to the empty set, which indicates that they do not retain offline data. Therefore, we implemented JSRL with no data retention.
>
> > RE: Added Experiments and Ablations
>
> In addition, We have also added new experiments showing WSRL’s performance on 1) nine mujoco locomotion environments as well as 2) eight antmaze environments in Appendix A and B. The experiments all follow the same trend as the ones in the main paper, and support our main claims.
>
> We also added new ablation experiments in Appendix D, E, I, J, further analyzing the design decisions we made in WSRL.
>
> > Minor Line 84: an warmup
>
> > Line 207: on which we measure metrics on
>
> > Line 253: onto the previous action pair that the TD error (?)
>
> > Figure 4: Mixed, complete, partial (what does this refer to?)
>
> Thanks for pointing them out, we have fixed them. In Figure 4, the “Mixed”, “Complete”, and “partial” refers to three different kinds of kitchen environments.
>
> [1] Nikishin, Evgenii, et al. "The primacy bias in deep reinforcement learning." International conference on machine learning. PMLR, 2022.
>
> [2] Uchendu, Ikechukwu, et al. "Jump-start reinforcement learning." International Conference on Machine Learning. PMLR, 2023. https://arxiv.org/abs/2204.02372

---

> > ### Comment · Reviewer_7N5M · 2024-11-23
> >
> > Thank you for your replies, and for adding in the additional experimentation in such as short time period. Thank you also for clarifying my misunderstanding in my questions and weaknesses. With regards to JSRL, they do specify that they used batches including 25% offline data in Appendix A.2.1, however as you pointed out, this is contrary to their description of the algorithm in Algorithm 1, so I think it is reasonable for you to use this implementation. I won't change my score as it is already an 8, it was a helpful paper for me as I had also observed this issue with not retaining offline data, but hadn't investigated it. I am glad the issue has been explained, and I was impressed with your level of experimentation.

---

> > > ### Author Response · Authors · 2024-11-24
> > >
> > > Thank you for your reply! We are glad to hear that you liked the paper and that our analysis and experiments help explain issues in no retention fine-tuning that you have also observed. Thanks for pointing out the inconsistency in the JSRL paper, we were not aware of this issue.

---

### Official Review · Reviewer_PY4G · 2024-11-01

**Soundness:** 3
**Presentation:** 2
**Contribution:** 3
**Rating:** 6
**Confidence:** 4

**Summary:**

This paper studies the Offline to Online fine-tuning problem in RL. The author trying to remove the need of keeping the offline dataset when finetuning online, which is becoming impractical giving the growing size of pretraining dataset. Through a detailed analysis of Q-values during the fine-tuning phase, the authors reveal two critical insights:
1. Without the offline dataset, pretrained Q-values are quickly unlearned
2. Continuing conservative offline RL on online data leads to performance degradation

In response, the authors propose WSRL, a method that:
1. Implements a warmup phase to seed the replay buffer with the pretrained policy to mitigate the unlearning issue
2. Utilizes state-of-the-art online RL algorithms for fine-tuning

The approach is demonstrated through promising results on D4RL tasks.

**Strengths:**

- This paper studies an important and timely topic in RL.
- This paper provides some analysis for transition phase to motivate the problems.
- The proposed method is simple to implement.

**Weaknesses:**

A lot of the results are confusing and not convincing, as referring in the questions below.

**Questions:**

- Could you clarify Figure 3? The caption says "Fine-tuning starts at step 0.", but the plots are start from around 250k.
- Some of the plots are truncated, i.e. not starting from 0. Could you please clarify on that? Based on the analysis, the warm-up phase should mostly solve the problem in the beginning of the finetuning phase. I would even suggest the authors to have a zoom-in version of that parts of the plot, but the truncated version seems to be missing exactly the transition part.
- Since warm-up phase is to just seed the replay buffer with some data collected from the pretrained policy, will it also be fine to subsample a small portion of the offline dataset to initialize the replay buffer?
- From the results in Figure 7, seem like the warmup phase does not really solve the unlearning problem, i.e. the initial value for WSRL is much lower than the CalQL initialization. Is this a desired behaviour?
- I am not sure how the warm-up phase solves the miscalibration problem present in CalQL. For example when you initialize the policy and Q with the CQL, although the warmup data will help to combat the distribution mismatch, the change of the RL algorithm will still make the Q function miscalibrated which in turn make the policy unlearn. How do you think of this issue?
- Won't it be beneficial to just run a bunch of gradient steps only on the Q function after you collect the warmup data until it recalibrates?

---

> ### Author Response · Authors · 2024-11-20
> **Paper updates and clarifications**
>
> Thank you for your feedback! We have addressed your comments by 1) adding zoomed-in plots for the start of fine-tuning, 2) adding analysis on why WSRL does not suffer from catastrophic forgetting of its offline pre-training, 3) answering your question on the warm-up phase solves the mis-calibration problem, and 4) adding ablations on only updating the Q function at the start of fine-tuning. We also added extensive experiments on nine Gym locomotion tasks and eight antmaze environments, as well as many ablations. Please let us know if any further concerns remain, or if these changes address the issues.
>
> > Some of the plots are truncated, i.e. not starting from 0. Could you please clarify on that? I would even suggest the authors to have a zoom-in version of that parts [the beginning of fine-tuning phase] of the plot.
>
> Thanks for the suggestion, we have added Figure 16 in Appendix D, showing the zoomed-in version of the first 50k steps of fine-tuning at small evaluation intervals. Some of the original plots did not start from 0 because the first performance evaluation and logging in our codebase was done $N$ steps into fine-tuning. Figure 16 shows performance starting from step 0. We also updated Figure 6 and 7 to start from step 0. These updates do not affect any of the conclusions in the paper.
>
> > From the results in Figure 7, seem like the warmup phase does not really solve the unlearning problem, i.e. the initial value for WSRL is much lower than the CalQL initialization. Is this a desired behaviour?
>
> You are right – Figure 7 shows that for Kitchen environments the initial value of WSRL is worse than CalQL, which in this comparison is provided access to retaining offline data. This is because, at the very early stage of fine-tuning, WSRL’s policy performance does take a little dip (this is better seen in the zoomed-in plots in Figure 16). However, Figure 16 also shows that WSRL is able to quickly recover from such a dip, and learns faster than baseline methods with no-retention. As Figure 7 shows, WSRL eventually learns a much better policy despite its initial dip; although CalQL has better initial performance, it never really improves much in Kitchen envs.
>
> We hypothesize that such a dip might be inevitable at the start of online fine-tuning in the no offline data retention setting because the policy is experiencing different states than what it was trained on and potentially states it has never seen (See Appendix D for an example and more details). Moreover, in some environments (e.g., binary reward environments), one might expect small fluctuations in the policy to manifest as large changes in the actual policy performance. However, such brief performance dip does not mean the policy/Q function has unlearned, which is evidenced by the fact that WSRL recovers faster than its peer algorithms and learns faster than online RL algorithms such as RLPD (See Figure 7). If this initial dip would have destroyed all pre-training knowledge from the policy, then we would not expect quick recovery.
>
> Therefore just deducing whether an algorithm has unlearned or not based on performance may not be the most informative. Instead, a more meaningful metric to measure unlearning is to evaluate how much a fine-tuning algorithm with no data retention deviates from its pre-training, and how fast it can adjust to the online state-action distribution. Therefore, we measure the KL divergence between the fine-tuned policy and the pre-trained policy on both online and offline data distributions, and find that WSRL remains stable during fine-tuning and does not destroy priors learned from offline pre-training. See a more detailed analysis in Appendix D.
>
> We acknowledge the term “unlearning” was not clearly defined in the paper, and claims of WSRL resolves the unlearning issue might have caused confusion. We have updated the paper to remove ambiguous claims of “unlearning”, and state that even though WSRL experiences an initial dip in policy performance, it quickly recovers and does not destroy priors learned from offline pre-training. We would appreciate any feedback you might have in making the terminology more clear in the paper.

---

> ### Author Response · Authors · 2024-11-20
> **Paper updates and clarifications**
>
> > I am not sure how the warm-up phase solves the miscalibration problem present in CalQL. For example when you initialize the policy and Q with the CQL, although the warmup data will help to combat the distribution mismatch, the change of the RL algorithm will still make the Q function miscalibrated which in turn make the policy unlearn. How do you think of this issue?
>
> Good question! As you mentioned, the change in RL algorithm from offline to online could lead to miscalibration issues, where the pre-trained values are more pessimistic than ground truth values. This miscalibration issue is amplified by the no-retention setting, as opposed to the problem setting in the CalQL paper where offline data is retained during fine-tuning. The reason warm-up mitigates this issue to a large extent is because, as you mention, warmup data helps combat distribution mismatch. We provide a detailed analysis of how this affects the Bellman update in Appendix F, and instead give a high-level intuition here. With a large number of training steps on this warmup data (i.e., with a high UTD) that is offline-like, we can obtain a better Q-function that retains pre-trained knowledge and successfully fine-tune. Note that while the no retention fine-tuning problem setting does not retain offline data, it allows as many updates needed on warmup data before more online data is collected and in the process, it can address this miscalibration issue by simply training on warmup data. Please let us know if this clarifies this question, we are happy to clarify further!
>
> > Won't it be beneficial to just run a bunch of gradient steps only on the Q function after you collect the warmup data until it recalibrates?
>
> Following your suggestion, we added a new ablation in Appendix E, showing the effect of freezing the policy and only updating the Q-function for some period of time after the warmup phase. As Figure 18 shows, this generally does not improve performance across six environments and two different freezing-periods, and does not resolve the initial dip in policy performance (again, more detailed discussion on why this is unavoidable in Appendix D).
>
> > RE: Added new experiments on diverse domains
>
> Additionally, We have also added new experiments showing WSRL’s performance on 1) nine mujoco locomotion environments as well as 2) eight antmaze environments in Appendix A and B. The experiments all follow the same trend as the ones in the main paper, and support our main claims.
>
> > Could you clarify Figure 3? The caption says "Fine-tuning starts at step 0.", but the plots are start from around 250k.
>
> Our apologies for the typo. The agents were pre-trained for 250k steps, and fine-tuning starts at step 250k.

---

> ### Comment · Reviewer_PY4G · 2024-11-21
>
> Thanks for your replies to my questions. Many concerns has been resolved by the rebuttal, except:
> - I can agree with you from Appendix D that somehow WSRL managed to forget fast and adapt fast and these add up together to make the final performance of WSRL better. But it is still unclear what does WSRL forget during the shift. And why is it only harmful for the short term (the performance dip) but not and even beneficial for the long term (the asymptotic performance). Like you said, the term "unlearning" needs to be more properly defined. I will encourage the authors to work more on this direction.
> - I think you tried to use Appendix I to answer my question about the buffer initialization instead of doing warmup. But that is not what I mean. What I mean is, as WSRL is doing, by default, 5000 steps for warmup with data collected by the pretrained policy, what if we instead randomly sample 5000 transitions from the pretrain offline dataset and use that for warmup. In this way, the method hopefully doesn't slowed down as you shown in Figure 20 with the entire offline dataset.
>
> Despite of these, I do think the new contents strengthen the paper. I will increase my score to reflect that.

---

> ### Author Response · Authors · 2024-11-23
>
> Thanks for your fast response! We are glad to hear that most of the concerns have been resolved. We wanted to provide some clarity on your last two questions. Please let us know if anything still remains unclear, we are very happy to discuss further.
>
> > what if we instead randomly sample 5000 transitions from the pretrain offline dataset and use that for warmup
>
> Thanks for the suggestion! We added a new experiment in Figure 22 for this ablation, along with analysis in Appendix K, highlighted in purple. The results show that using 5000 transitions from the offline dataset for warmup generally performs worse than WSRL. Note that this experiment highlights the utility of our approach: despite not having access to _any_ offline data, WSRL is able to achieve similar or better performance than using transitions from the offline dataset in the no retention fine-tuning setting.
>
> To dive into the offline dataset result in more detail, while the two methods are similar on Adroit environments, WSRL is slightly better in Kitchen and much better on Antmaze. This is perhaps because the 5000 randomly sampled transitions might not be relevant to the online fine-tuning task, especially in Antmaze where the dataset has diverse state-action coverage (as explained below and in Appendix D). When the replay buffer is initialized with less relevant data, it is less effective at preventing Q-value divergence (Section 4) and recalibrating the online Q-function and policy.
>
> > But it is still unclear what does WSRL forget during the shift. And why is it only harmful for the short term (the performance dip) but not and even beneficial for the long term (the asymptotic performance).
>
> That’s a great question! We added more analysis to Appendix D (highlighted in purple) to further aim to understand this, though of course just studying unlearning in all can be a full paper in itself and we do not mean to imply that we provide an extensive analysis studying this.
>
> In short, our experiments highlight that due to the distribution shift from offline pre-training to no-retention fine-tuning, WSRL is forgetting experience in the offline dataset that was learned during offline pre-training but is in reality irrelevant for specializing to the online task. In contrast, it is instead specializing to the online task. These intuitions of “forgetting” and “specializing” can be seen in the two KL divergence plots that we have added in Figure 17:  concretely, these plots show the KL divergence between the offline policy and the fine-tuned policy on offline and online state distributions. The KL divergence increases over the offline state distribution, which suggests that the policy learns to *forget* at least some behaviors appearing in the offline dataset that it had learned during offline pre-training; but the KL divergence remains small on the online state distribution, suggesting that the policy did not forget the behavior it had learned from the offline data relevant to the online distribution. We do note the caveat that KL divergence may not be entirely indicative of forgetting and plan to continue to investigate the broader question in our future research.
>
> That said, to give some specific examples, consider the Antmaze environments and the Adroit environments. The Antmaze offline dataset exhibits a very diverse state-action distribution, covering almost all the locations in the entire maze. However, the online task is single-goal navigation. Therefore, in no-retention fine-tuning, the agent is incentivized to *forget* irrelevant states (because it is no longer training on those experiences) and instead only specialize in the fine-tuning task. Therefore, the KL divergence on the offline state distribution increases. However, the KL divergence on the online distribution remains small because the pre-trained policy is somewhat capable on such states and the policy only needs to adjust to some unseen states.
> On the other hand, the Adroit datasets consist of expert human demonstrations and rollouts of a good behavior cloning policy. One can imagine that these experiences are all relevant to solving the same task as what we wish to solve during fine-tuning. As expected, we see that the KL divergence on offline distribution on Adroit environments barely increases (as opposed to Kitchen and Antmaze), suggesting that the policy did not forget useful priors in pre-training.

---

> > ### Comment · Reviewer_PY4G · 2024-11-26
> >
> > Thanks for your new experiments regarding seeding the replay buffer with the offline data. It does convince me that it is a better idea to seed the buffer with the pretrained policy.
> >
> > Regarding the explanation of the "unlearning", it is a nice hypophysis, but I am afraid it is not well-aligned with the results.
> > - Besides the AntMaze, other tasks' offline goal is aligned with the online goal. Whether it does it offline or online, they are optimising the same reward.
> > - The KL plots in Figure 17 are truncated as before. Since the online policy is initialised with the pretrained policy, all these curves should start at 0. If we are using KL to quantify how much knowledge does the policy forget, it is opposite to your explanation -- it forgets most on the Adroit tasks.

---

> > > ### Author Response · Authors · 2024-11-27
> > > **Further clarifications**
> > >
> > > Thanks for your response! Glad to hear that we were able to answer your question regarding seeding the replay buffer with the offline data, and justify the value of our approach of warmstarts, which is the main contribution of the paper. To answer your other questions:
> > >
> > > > Besides the AntMaze, other tasks' offline goal is aligned with the online goal. Whether it does it offline or online, they are optimising the same reward.
> > >
> > > You are correct, these environments optimize the same reward function (and therefore task) offline and online. However, online training could still differ significantly from offline training. The primary difference lies in the composition / distribution of the offline dataset, which determine the state-action pairs used to constrain the policy through offline RL and the state-action pairs on which updates are done. Such differences could determine how fast the pre-trained policy can adapt to the online distribution, and how much the pre-trained policy unlearns.
> > >
> > > For example, as you mentioned, the AntMaze datasets contain significantly more information than the online task and the policy is trained on these states and constrained to this behavior. In the Kitchen environments, the kitchen-partial dataset includes extra tasks not present during fine-tuning, while the kitchen-mixed dataset contains only subtasks and no full positive trajectories. This may require the agent to “forget” extra information in kitchen-partial. For Adroit, the dataset consists of demonstrations and closely resembles the online task. However, we suspect that in this particular setting, its small size can lead to insufficient offline RL pre-training such that it might be preferable for online fine-tuning to not stay close to the offline policy, potentially making the agent “forget” more easily (for all the methods under consideration: WSRL, CalQL, and CQL). That said, we do admit that these are all hypotheses and we hope our observations provide inspiration for future work and potential explanation of the phenomena we see.
> > >
> > > > If we are using KL to quantify how much knowledge does the policy forget, it is opposite to your explanation -- it forgets most on the Adroit tasks.
> > >
> > > You are correct in that the KL divergence is high on Adroit, and we apologize for the confusion in Appendix D. Despite the high KL divergence, we note that all methods attain the same asymptotic value of KL divergence on this domain. This likely implies that the magnitude of the KL divergence on a given domain should also be a function of the offline dataset and the online task, and may not be comparable across tasks. We have removed the comparisons of KL divergence across tasks from Appendix D. Regardless, Figure 17 shows that WSRL “adjusts to the online distribution” much faster than CQL and CalQL (shown by how fast the KL divergence reaches the asymptotic value), perhaps thanks to its non-conservative objective optimized solely on online data and its high update-to-data ratio during online RL.
> > >
> > > That said, we do admit that there is more room for improvement in explaining the “unlearning” phenomenon. We are not claiming that we fully explain unlearning or forgetting in this paper, and will add this discussion as a limitation and an inspiration for future work. We are still working on running some experiments to understand the degree of unlearning in the policy vs. the Q-function, and whether the offline pre-training is good enough to learn effective initializations that are representative of the dataset distribution. Due to time constraints, these experiments may have to be added after the revision period ends, though please rest assured that we will add this discussion since all of this has improved the value of the paper! We would also welcome any suggestions you might have in understanding what and why WSRL is unlearning.

---

### Official Review · Reviewer_tchB · 2024-11-02

**Soundness:** 2
**Presentation:** 2
**Contribution:** 2
**Rating:** 6
**Confidence:** 4

**Summary:**

The paper proposes a new method, Warm Start Reinforcement Learning (WSRL), for efficient online reinforcement learning (RL) fine-tuning without the need to retain offline data. Traditional RL fine-tuning often relies on retaining offline data to prevent performance degradation, but this approach is inefficient and can slow down learning. The authors argue that, by using a warmup phase that gathers initial online rollouts with a pre-trained policy, WSRL can "recalibrate" the offline Q-function to online data, enabling the algorithm to discard offline data and still achieve stable and effective learning. WSRL, which emphasizes a high updates-to-data (UTD) regime, proves faster and more effective across diverse tasks compared to traditional algorithms, demonstrating higher performance without offline data retention.

**Strengths:**

- WSRL removes the dependency on large offline datasets for online fine-tuning, based on various experimental takeaways.
- WSRL shows improved learning speed and higher asymptotic performance across various benchmark tasks, outperforming state-of-the-art fine-tuning methods.
- Introducing a warmup phase and applying a high UTD regime for online RL is an efficient approach to address the problem of Q-function divergence without retaining offline data.
- The method is adaptable across different RL environments, showing versatility and potential applicability to a broad range of tasks.

**Weaknesses:**

- The offline performance in the experimental setup is poor from the start (Section 6, Figures 6 and 7). The performance graph shows that, except for pen-binary, the offline score ranges from 0 to 20 points, which is quite low.  This weak offline performance may suggest that the offline data did not fit as effectively as it does with other algorithms or that, in this particular setup, the offline data may not have provided much support for online fine-tuning. The claim that the proposed method prevents unlearning is also debatable, as there was minimal offline performance to “unlearn” initially. Analyzing cases where the offline score reaches at least 60, ideally between 80 and 100, would be more informative. Conducting further experiments on Mujoco Hopper/Walker2D with medium, medium-replay, expert, or Adroit-dense expert settings—where offline performance tends to be stronger—would be beneficial.
  - Additionally, this paper mentions using SAC with a 10-ensemble for training, but it would also be necessary to examine how other offline algorithms like IQL and CQL perform when applying the proposed methods—such as excluding offline data, using a warmup phase, and employing a high UTD ratio.

- Unlearning likely occurs mostly at the very early stages of training, but the plots are spaced at intervals of about 50,000 steps. At this interval, even if unlearning did occur, it may have already been resolved by the next plotted point. To demonstrate the absence of unlearning, it would help to show early performance at smaller timestep intervals.

- The paper employs strong language, such as “Should Not” and “Completely unnecessary,” though the evidence provided may not be fully convincing. Without theoretical support, it needs stronger empirical backing; however, the experiments may not comprehensively represent all RL tasks. Removing such strong expressions would not alter the flow of the paper and might actually improve its persuasiveness.

- This paper conducted experiments on eight datasets across three domains (Antmaze ultra-diverse/large-diverse/large-play, Kitchen partial/mixed/complete, and Adroit door/pen), which appears insufficient compared to other conference papers. While experiments were conducted on Antmaze ultra-diverse, ultra-play was not included, which may suggest some inconsistency in selection. Given that this paper is experiment-driven and lacks a strong theoretical foundation, it would benefit from more extensive experiments, such as including Antmaze ultra/large/medium/umaze play/diverse datasets, to provide a more comprehensive set of results.

- In Figure 4 of Section 4.1 and Figure 9 of Section 6.5, it’s unclear what the Q-value magnitude itself indicates. Without a comparison to the true Q-value, it could be that one is underestimated or, conversely, that the other is overestimated. Additionally, even if the overall scale is low or high, as long as the general trend follows the true Q, there is no issue (the relative comparison of Q values is more important than their absolute magnitude). For a meaningful comparison, it might be better to separate in-distribution Q from out-of-distribution Q. t. Additionally, considering the focus on Q-value analysis and the approach from a Q-value perspective, I am curious to see a comparison with the paper “A Perspective of Q-Value Estimation on Offline-to-Online Reinforcement Learning” [1].

Overall, while achieving better performance than the baselines in the experimental settings (especially for Antmaze Large and Ultra) is highly positive, the points discussed above prevent the claim that "Efficient Online Reinforcement Learning Fine-Tuning Should Not Retain Offline Data" from being fully convincing.

[1] Zhang, Yinmin, et al. "A Perspective of Q-value Estimation on Offline-to-Online Reinforcement Learning." Proceedings of the AAAI Conference on Artificial Intelligence. Vol. 38. No. 15. 2024.

**Questions:**

- Could you provide experimental results from environments like the MuJoCo dataset, where offline performance is typically strong?
- Could you provide experimental results using other offline algorithms (e.g., IQL, CQL) with the proposed methods applied?
- Could you provide experimental results for all 8 Antmaze datasets?
- Could you provide a comparison with “A Perspective of Q-Value Estimation on Offline-to-Online Reinforcement Learning”?
- This paper excludes offline data during online fine-tuning, implements a warmup phase, and increases the UTD ratio. What would happen if we retained the offline data while implementing the warmup phase and increasing the UTD ratio?

---

> ### Author Response · Authors · 2024-11-20
> **Paper updates and clarifications**
>
> Thank you for your feedback! We have addressed your comments by 1) added experiments on all eight antmaze environments and nine Mujoco locomotion environments, 2) including baseline comparison against SO2 agent from Zhang et al., which we find still underperforms our method WSRL, 3) added zoomed-in plots for start of fine-tuning, 4) clarified on offline pre-training performance and added analysis on whether WSRL “unlearns”, and 5) added many ablation experiments such as IQL/CQL pre-training and WSRL with data retention. Please let us know if any further concerns remain, or if these changes address the issues.
>
> > Could you provide experimental results from environments like the MuJoCo dataset?  Could you provide experimental results for all 8 Antmaze datasets?
>
> We have added results for WSRL (along with baselines) on nine Mujoco locomotion environments in Appendix B, as well as results on all eight Antmaze environments in Appendix A. The trends observed in these environments are consistent with those presented in the submission: WSRL outperforms previous state-of-the-art algorithms, such as CalQL and RLPD, despite some baselines benefiting from the unfair advantage of retaining offline data (See Figure 12, 13). We believe these additional experiments should address your concern on comprehensive evaluation of WSRL on diverse domains.
>
> > Could you provide a comparison with “A Perspective of Q-Value Estimation on Offline-to-Online Reinforcement Learning”?
>
> Thanks for pointing out this work. We have added this work as a baseline in Figure 7. The results show WSRL performing similarly on Adroit environments, but much better in Kitchen and Antmaze environments.
>
> > While experiments were conducted on Antmaze ultra-diverse, ultra-play was not included, which may suggest some inconsistency in [dataset] selection.
>
> We picked the three antmaze environments out of the eight because they were three of the hardest antmaze environments to solve, as measured by performance of previous offline RL and online RL fine-tuning algorithms (CalQL [2], Proto [3], IQL [4]). We only presented results on antmaze-ultra-diverse and not antmaze-ultra-play because their dataset collection mechanism is very similar. However, we agree with you that the paper would benefit from more comprehensive results, so we added experiments for all eight antmaze environments in Appendix A (See Figure 12). Across the board, we find that WSRL outperforms other methods that do retain offline data.
>
> > The offline performance in the experimental setup is poor from the start (Section 6, Figures 6 and 7)
>
> Sorry for the confusion. The offline performance on the environments in the paper actually is good (See updated Figures 6 and 7, first data point), some reaching 60-80 points, as you mentioned. The confusion stems from the fact that the original Figures 6 and 7 were plotted at large evaluation intervals, and the first evaluation data point did not reflect the offline performance. We have fixed Figures 6 and 7 to show offline performance as the first data point.

---

> ### Author Response · Authors · 2024-11-20
> **Paper updates and clarifications**
>
> > To demonstrate the absence of unlearning, it would help to show early performance at smaller timestep intervals
>
> This is a great suggestion! To address this question, we have now added a new result in Figure 16 in Appendix, showing the fine-tuning performance of WSRL and baseline algorithms during the first 50k steps with very small evaluation intervals.
>
> Figure 16 shows that WSRL experiences an initial dip in policy performance after 5, 000 steps of warmup, but recovers much faster than other baselines. We hypothesize that such a dip might be inevitable at the start of online fine-tuning in the no offline data retention setting because the policy is experiencing different states than what it was trained on and potentially states it has never seen (See Appendix D for an example and more details). Moreover, in some environments (e.g., binary reward environments), one might expect small fluctuations in the policy to manifest as large changes in the actual policy performance. However, such brief performance dip does not mean the policy/Q function has unlearned, which is evidenced by the fact that WSRL recovers faster than its peer algorithms and learns faster than online RL algorithms such as RLPD (See Figure 7). If this initial dip would have destroyed all pre-training knowledge from the policy, then we would not expect quick recovery.
>
> Therefore just deducing whether an algorithm has unlearned or not based on performance may not be the most informative. Instead, a more meaningful metric to measure unlearning is to evaluate how much a fine-tuning algorithm with no data retention deviates from its pre-training, and how fast it can adjust to the online state-action distribution. Therefore, we measure the KL divergence between the fine-tuned policy and the pre-trained policy on both online and offline data distributions, and find that WSRL remains stable during fine-tuning and does not destroy priors learned from offline pre-training. See a more detailed analysis in Appendix D.
>
> We acknowledge the term “unlearning” was not clearly defined in the paper, and claims of WSRL resolves the unlearning issue might have caused confusion. We have updated the paper to remove ambiguous claims of “unlearning”, and state that even though WSRL experiences an initial dip in policy performance, it quickly recovers and does not destroy priors learned from offline pre-training. We would appreciate any feedback you might have in making the terminology more clear in the paper.
>
> > Could you provide experimental results using other offline algorithms (e.g., IQL, CQL) with the proposed methods applied?
>
> In the submission PDF, we show in Figure 19 in Appendix G that WSRL performs similarly across three environments regardless of whether we use IQL, CQL, CalQL for the offline pre-training. We are working on obtaining similar results in more environments.
>
> > The paper employs strong language, such as “Should Not” and “Completely unnecessary,” though the evidence provided may not be fully convincing.
>
> Thank you for the suggestion! Following your suggestion, we have removed strong language from the paper, such as “Should Not” and “completely unnecessary”. For example, we propose to change the title to “Efficient Online Reinforcement Learning Fine-Tuning ***Need Not** Retain Offline Data”, instead of the original title. We have also added many experiments in additional environments (Figure 12, 13) to support our claims. All changes in the paper are highlighted in blue. Please let us know if there are other instances of strong claims that you would want us to change, and we are happy to make changes!
>
> > This paper excludes offline data during online fine-tuning, implements a warmup phase, and increases the UTD ratio. What would happen if we retained the offline data while implementing the warmup phase and increasing the UTD ratio?
>
> We added an additional ablation in Figure 20 in Appendix I to compare performance of WSRL vs. WSRL with the replay buffer initialized with offline data. We find that the performance is mostly similar, though initializing with offline data sometimes provides stronger performance. This is not surprising and perhaps expected because in certain environments, retaining the offline dataset could give the fine-tuning more information, possibly information that was not well-learned during pre-training. That said, it does not conflict with our goal in this paper which is to build a method without data retention that can outperform existing fine-tuning approaches which seem to entirely fail when offline data is not retained (as we show in Figure 6), and WSRL does indeed satisfy this desideratum, even if WSRL + offline data performs similarly to WSRL.

---

> ### Author Response · Authors · 2024-11-20
> **References**
>
> [1] Zhang, Yinmin, et al. "A Perspective of Q-value Estimation on Offline-to-Online Reinforcement Learning." Proceedings of the AAAI Conference on Artificial Intelligence. Vol. 38. No. 15. 2024.
>
> [2] Nakamoto, Mitsuhiko, et al. "Cal-ql: Calibrated offline rl pre-training for efficient online fine-tuning." Advances in Neural Information Processing Systems 36 (2024).
>
> [3] Li, Jianxiong, et al. "Proto: Iterative policy regularized offline-to-online reinforcement learning." arXiv preprint arXiv:2305.15669 (2023).
>
> [4] Kostrikov, Ilya, Ashvin Nair, and Sergey Levine. "Offline reinforcement learning with implicit q-learning." arXiv preprint arXiv:2110.06169 (2021).

---

> > ### Comment · Reviewer_tchB · 2024-11-23
> >
> > Thank you for your response. I'm impressed by the amount of experimentation and clarification you provided in such a short time. Most of my concerns have been resolved, so for now, I've raised my rating to 5. I'm open to increasing it further, but I haven't yet had the chance to thoroughly analyze the revised content, so I haven't fully organized my thoughts yet. I might ask a few more follow-up questions, or if I find the current content sufficient, I might simply raise the rating further. Since I was the only reviewer who gave a reject rating, I thought you might be curious about my opinion, so I decided to share my thoughts before it’s too late. I hope this doesn't cause any confusion, as my feedback isn't fully organized yet.

---

> > > ### Author Response · Authors · 2024-11-23
> > >
> > > Thanks for your fast response and your affirmation of our efforts during this period. We are glad to hear that most of the concerns have been resolved! Please let us know if anything still remains unclear, we are happy to discuss further.

---

> ### Comment · Reviewer_tchB · 2024-11-25
> **Additional question for clarification**
>
> Thank you for addressing the concerns with additional experiments, as noted in the previous response. However, there are still unclear aspects, particularly in the writing and presentation of experimental results.
>
> The paper's purpose and contributions, as I understand them, are as follows:
>
> **“In offline-to-online RL, retaining offline data during online fine-tuning is impractical. Therefore, the paper proposes a novel method to perform well in online fine-tuning without retaining offline data, supported by various analyses and a new warm-up phase.”**
>
> This premise is reasonable and represents a meaningful contribution.
>
> However, there appear to be some areas where the content of the paper and the experimental results are not fully aligned.
>
> &nbsp;
>
> **1. Inconsistent Use of "Unlearning" and "Catastrophically Forgetting"**
>
> From my understanding:
>
> - **"Unlearning"** in this paper refers to the performance drop at the start of online fine-tuning due to distribution shift.
> - **"Catastrophically forgetting"** refers to information loss so severe that recovery becomes impossible even with continued training.
>
> The terms are used in the following contexts in the paper:
>
> **<Unlearning>**
>
> - Line 20: "We find that continued training on offline data is mostly useful for preventing a sudden unlearning of the offline RL value function at the onset of fine-tuning, caused by a distribution mismatch between the offline data and online rollouts."
> - Line 23: "Our approach, WSRL, mitigates this sudden unlearning..."
> - Line 73: "This recalibration phase can lead to unlearning of the offline initialization, and even divergence, when no offline data is present for training."
> - Line 79: "We show that the main culprit behind the unlearning is the distribution mismatch between the offline data and online training distribution, and retaining offline data attenuates the effect of this mismatch, playing an essential role in the working of current offline-to-online fine-tuning methods."
>
> **<Catastrophically Forgetting>**
>
> - Line 82: "...catastrophically forgetting offline pre-training."
> - Line 85: "'Simulates' offline data retention can greatly facilitate recalibration, preventing catastrophic forgetting that never recovers with more learning."
> - Line 325: "How can we tackle both catastrophic forgetting of the offline initialization and attain asymptotic sample efficiency online?"
> - Line 330: "The remaining question is: how do we tackle catastrophic forgetting at the onset of fine-tuning that prevents further improvements online, without offline data?"
>
> As explained in the revised Appendix D (the initial performance dip might be inevitable due to the interaction of the learning policy with new states not present in the pretraining data), the paper does not resolve **unlearning** but rather addresses **catastrophic forgetting**. This distinction must be clarified explicitly in the paper without any confusion. Claims that the paper resolves **unlearning** are incorrect and misleading.
>
> &nbsp;
>
> **2. Step 0 Discrepancy in Figures 6 and 7**
>
> - In Figures 6 and 7, the values for WSRL at step 0 differ. Is there an explanation for this? The trends appear identical after step 0, making this discrepancy seem unusual.
>
> &nbsp;
>
> **3. Details on MuJoCo Experiments**
>
> - Thank you for conducting additional MuJoCo experiments. However, the details of these experiments are not provided. Were they conducted under the same settings as described in Line 1065 (hyperparameter section)?
>
> &nbsp;
>
> **4. Minor**
>
> - Y-Axis Format in Figures 3 and 4: The y-axis uses exponential notation (e.g., 10^{−2.1}) in decimal form. Is there a specific reason for this choice?
> - Consistency in Terminology: The term "Cal-QL" is written both as "CalQL" and "Cal-QL." Please standardize its usage throughout the paper.
> - Dataset Order in Figures: In Figures 6 and 7, the order of datasets differs. Unless there is a specific reason, aligning the order would improve clarity.
>
> &nbsp;
>
> I believe the contribution is valuable and the experiments are thorough, but it’s important for the claims in the paper to align with the actual findings to maintain clarity and accuracy. It would be helpful if the authors could indicate whether the writing can be improved, even if immediate revisions are not possible due to the limited rebuttal period. Alternatively, if my understanding is mistaken and the current writing already accurately reflects the experimental results, I would appreciate clarification.

---

> > ### Author Response · Authors · 2024-11-26
> > **Further Clarifications**
> >
> > Thanks for your response and your detailed feedback! We are glad to hear you found the contribution valuable and the experiments thorough. We have improved the writing to clearly define **unlearning** and **catastrophic forgetting** in **Section 4.1**, and carefully state that WSRL only prevents the latter in **Section 6.3** (edits highlighted in blue, keywords in **bolded text**). We believe the new clarifications should clear up the confusion. We are also actively working on including experiments to analyze the unlearning phenomenon in WSRL (in addition to the ones we already added during the rebuttal). Due to time constraints they may have to be added after the revision period ends, though please rest assured that we will add them since they improve the value of the paper! Please let us know if you have any other concerns in accepting the paper, and we would be very happy to discuss.
> >
> > > In Figures 6 and 7, the values for WSRL at step 0 differ. Is there an explanation for this? The trends appear identical after step 0, making this discrepancy seem unusual.
> >
> > Thank you for pointing this out. The discrepancy at step 0 arose because, during the rebuttal process, we added step 0 values using different random seeds for Figures 6 and 7 due to a collaborative effort to include step 0 results. While the values are expected within error margins, they differ slightly and may cause confusion, as you pointed out. To address this, we have updated the plots using a consistent set of seeds, ensuring WSRL values are identical in Figures 6 and 7.
> >
> > > Thank you for conducting additional MuJoCo experiments. However, the details of these experiments are not provided. Were they conducted under the same settings as described in Line 1065 (hyperparameter section)?
> >
> > Yes, the Mujoco experiments have the same setting as in Line 1065. We have added further experiment details in Appendix B.
> >
> > > Y-Axis Format in Figures 3 and 4: The y-axis uses exponential notation (e.g., 10^{−2.1}) in decimal form. Is there a specific reason for this choice?
> >
> > We chose the Y-axis scale so that it clearly displays the trends of the three or four lines, ensuring each line is distinctly visible and without significant overlap. Just as an example, if the y-axis is not in exponential form, Figure 3(b) would look like [this](https://postimg.cc/mzwwDHpj), with significant white space and overlap between the lines.
> >
> > > Consistency in Terminology: The term "Cal-QL" is written both as "CalQL" and "Cal-QL." Please standardize its usage throughout the paper.
> >
> > Thanks for pointing this out. We have standardized to “CalQL” in the paper.
> >
> > > Dataset Order in Figures: In Figures 6 and 7, the order of datasets differs. Unless there is a specific reason, aligning the order would improve clarity.
> >
> > Thanks for pointing this out. We have updated the two figures to align the ordering.
> >
> > **Please let us know if these clarifications and the paper revisions address these concerns. We are happy to address any more concerns or answer any more questions, thanks for the great feedback – it has been very helpful!**

---

> > > ### Comment · Reviewer_tchB · 2024-11-27
> > >
> > > Thank you for your response. Many issues have now been addressed, but since I first read this paper, I have felt that it had a tendency to overclaim in some areas. I also believe there is still room for improvement in the writing. Nevertheless, I think the claims themselves are sufficient to make a sufficient contribution, so I am raising the score to 6. However, if a paper with overclaims is published, there is a risk that the community might mistakenly assume that other unresolved problems have also been solved. I hope this issue can be clearly addressed in the camera-ready version, as this paper could serve as a benchmark or starting point for other papers.

---

> > > > ### Author Response · Authors · 2024-11-27
> > > >
> > > > Thank you for your affirmation of our contribution and for raising your score! We appreciate the discussion and all your suggestions during these two weeks, which has helped greatly improve the paper. We agree that the writing should be refined and we will work on it for the camera ready. Thank you so much for helping catch wording that was confusing and gave an impression of overclaim, and we will make sure to pay special attention to wording and precision for the next revision.

---

### Official Review · Reviewer_Wxna · 2024-11-10

**Soundness:** 3
**Presentation:** 4
**Contribution:** 3
**Rating:** 6
**Confidence:** 3

**Summary:**

This paper explores the potential for efficient offline-to-online RL without retaining offline data. The authors investigate how offline data contributes to online fine-tuning in existing methods and propose WSRL, which leverages the offline policy and Q-function as initialization. Extensive experiments highlight the effectiveness of WSRL and its components.

**Strengths:**

This paper is well-written and easy to follow, tackling the interesting topic of offline-to-online RL. The study of related works is thorough and comprehensive. Building on this foundation, the paper illustrates how retaining offline data contributes to previous approaches and provides three key takeaway messages. The authors then propose WSRL, a simple and intuitive method designed to enhance the stability and efficiency of online fine-tuning.

The experimental results show that WSRL outperforms or is at least competitive with previous methods. The ablation studies are detailed and convincingly demonstrate the necessity of all components.

**Weaknesses:**

I did not identify any major flaws in this work. One potential limitation, however, is that WSRL relies on off-policy online algorithms, as it requires interactions from the warmup phase to stabilize the fine-tuning process.



Typos:

Ofline --> Offline (In the box of Takeway 3, Line 3)

**Questions:**

While off-policy algorithms are naturally suited for the online phase, is it possible to apply online policy methods with monotonic improvement properties (e.g., PPO, TRPO) and still achieve comparable performance during online training?

---

> ### Author Response · Authors · 2024-11-20
> **Paper updates and clarifications**
>
> Thank you for your time and your review! We answer below your question about WSRL’s compatibility with on-policy RL algorithms. We also added extensive experiments on nine Gym locomotion tasks and eight antmaze environments, as well as many ablations. Are there additional questions we could address to improve the score?
>
> > While off-policy algorithms are naturally suited for the online phase, is it possible to apply online policy methods with monotonic improvement properties (e.g., PPO, TRPO) and still achieve comparable performance during online training?
>
> That’s a great question! In this paper, we choose off-policy algorithms for fine-tuning because they are most compatible with the standard offline RL pre-training (e.g. CQL, IQL), where off-policy algorithms are necessary to learn from a pre-collected dataset. With off-policy pre-training, using the same type of learning objective online minimizes catastrophic forgetting. However, we do agree that on-policy fine-tuning from off-policy pre-training could be an interesting and important area for future work.
>
> > RE: new experiments and ablations
>
> We have also added new experiments showing WSRL’s performance on 1) nine Gym locomotion environments as well as 2) eight antmaze environments in Appendix A and B. The experiments all follow the same trend as the ones in the main paper, and support and strengthen our main claims.
>
> We also added new ablation experiments and analysis in Appendix D, E, I, J, further analyzing the design decisions we made in WSRL.
>
> > Typos: Ofline --> Offline (In the box of Takeway 3, Line 3)
>
> Thanks for pointing it out, we have corrected it.

---

> > ### Comment · Reviewer_Wxna · 2024-11-25
> >
> > Thank you for the thoughtful responses. I also reviewed the comments and discussions with other reviewers, particularly those with **Reviewer tchB**. I believe the rebuttal has significantly strengthened the work, and the change in the title from "Should Not" to "Need Not" is a thoughtful and more accurate adjustment.
> >
> > While the experiments convincingly demonstrate the effectiveness of the warm-up phase, I still find it challenging to develop an intuitive explanation for why it performs so well. Given the timing of my reply, there may not be sufficient opportunity to include a comparison between the data collected during the warm-up phase and the offline data, but such an analysis could potentially prove valuable for developing other approaches, whether or not they follow a data retention paradigm.
> >
> > Based on the improvements and the clarifications provided, I would like to increase my score by 1 (very slightly below a score of 8).

---

> > > ### Author Response · Authors · 2024-11-26
> > >
> > > Thank you for your response, and we appreciate your score increase from 6 to 7.
> > >
> > > We agree that it’s helpful to include a comparison between the warm-up data and the offline data. We will include a study of this in the appendix, though as you mentioned, it may be after the rebuttal period because of time constraints.

---

### Author Response · Authors · 2024-11-20
**Short summary of paper updates**

Thank you to all the reviewers for your feedback and comments! We have updated the paper (edits highlighted in blue) with many additional experiments, ablations, and clarifications. In summary, we have (1) added new experiments on nine Mujoco locomotion domains and all eight D4RL antmazes, and find that WSRL performs well compared to baselines, (2) added another recent baseline method from Zhang et al. [1], and found that it underperforms WSRL, (3) clarified on the fine-tuning performance of WSRL at the very start of fine-tuning and added analysis of whether it has lost its pre-trained knowledge, and (4) added many ablations on design decisions of WSRL.

Please let us know if you have any additional concerns and we are happy to discuss!

[1] Zhang, Yinmin, et al. "A Perspective of Q-value Estimation on Offline-to-Online Reinforcement Learning." Proceedings of the AAAI Conference on Artificial Intelligence. Vol. 38. No. 15. 2024.

---

### Meta-Review · Area_Chair_cD8X · 2024-12-15

**Metareview:**

This paper addresses the issue of retaining offline data during the fine-tuning phase of reinforcement learning (RL), which is often inefficient and impractical with large datasets. The authors propose a method called Warm Start Reinforcement Learning (WSRL), which eliminates the need for offline data retention by introducing a warmup phase that uses a small number of rollouts from a pre-trained policy to recalibrate the offline Q-function. This approach stabilizes the online fine-tuning process and improves performance without requiring offline data, demonstrating faster learning and higher performance compared to traditional RL fine-tuning methods.

The strengths of this paper include its contribution of an efficient approach and better understanding of the problem, along with competitive experimental results. The main weaknesses are some missing justifications and claims that need clarification, though the authors have addressed many of these and promised to conduct further clarifications.

The four reviews are all positive (6, 6, 6 [actually 7], and 8), suggesting that this paper should be accepted.

**Additional Comments On Reviewer Discussion:**

Reviewer tchB asked for several clarifications and rephrasing of overclaims. During the discussion, the authors made sufficient changes and promised more in the camera-ready version. Reviewer tchB increased their rating from negative to positive.

Reviewer Wxna had no major issues and increased their rating based on rebuttals for other reviewers. Wxna also encouraged extra analysis of the collected data, which the authors promised to include.

Reviewer PY4G had several questions about the results. While the majority of them have been addressed by the authors, PY4G increased their rating.

Additionally, it is worth noting that the title changed from "Should Not" to "Need Not" during the revision.

---

### Decision · Program_Chairs · 2025-01-22

Accept (Poster)